# Ultrafast light-activated polymeric nanomotors

Jianhong Wang[1,6], Hanglong Wu [1,6], Xiaowei Zhu [2], Robby Zwolsman[3], Stijn R. J. Hofstraat [3], Yudong Li [1], Yingtong Luo[1], Rick R. M. Joosten[4], Heiner Friedrich [4], Shoupeng Cao[5], Loai K. E. A. Abdelmohsen [1], Jingxin Shao [1] ✉ & Jan C. M. van Hest [1] ✉

Synthetic micro/nanomotors have been extensively exploited over the past decade to achieve active transportation. This interest is a result of their broad range of potential applications, from environmental remediation to nanomedicine. Nevertheless, it still remains a challenge to build a fast-moving biodegradable polymeric nanomotor. Here we present a light-propelled nanomotor by introducing gold nanoparticles (Au NP) onto biodegradable bowl-shaped polymersomes (stomatocytes) via electrostatic and hydrogen bond interactions. These biodegradable nanomotors show controllable motion and remarkable velocities of up to 125 µm s$^{-1}$. This unique behavior is explained via a thorough three-dimensional characterization of the nanomotor, particularly the size and the spatial distribution of Au NP, with cryogenic transmission electron microscopy (cryo-TEM) and cryo-electron tomography (cryo-ET). Our in-depth quantitative 3D analysis reveals that the motile features of these nanomotors are caused by the nonuniform distribution of Au NPs on the outer surface of the stomatocyte along the z-axial direction. Their excellent motile features are exploited for active cargo delivery into living cells. This study provides a new approach to develop robust, biodegradable soft nanomotors with application potential in biomedicine.

Self-propelled autonomous nanomotors are capable of converting energy into mechanical movement and have witnessed increasing attention owing to their large potential in biomedical applications[1–6]. Compared to traditional nanocarrier systems (without propulsive features), active particles can engage more effectively with cells and can directly translocate across the cell membrane[7–9]. Over the past decade, a variety of nanomotors has been designed, such as light propelled nanomotors[10–13], bubble driven nanomotors[14–16], and glucose assisted bi-metallic nanomotors[17]. Such miniaturized machines are

propelled by chemical fuel (e.g., hydrogen peroxide and urea)[18–21] or external stimuli (e.g., magnetic, ultrasound and light)[22–25]. The most representative driving forces and maximum velocities are summarized in Supplementary Table 1, from which it can be concluded that the highest velocities, up to 86 µm s$^{-1}$ are usually reached by nanomotors driven by thermophoretic forces, using light as energy source[26]. Nanomotors with an ultrafast velocity could lead to better tissue penetration, more efficient intracellular drug delivery, and enhanced accumulation in the target area[27–29]. Furthermore, because of the high

[1]Bio-Organic Chemistry, Departments of Biomedical Engineering and Chemical Engineering & Chemistry, Institute for Complex Molecular Systems, Eindhoven University of Technology, 5600 MB Eindhoven, The Netherlands. [2]School of Aeronautic Science and Engineering, Beihang University, Beijing 100191, China. [3]Laboratory of Chemical Biology, Department of Biomedical Engineering, Eindhoven University of Technology, 5600 MB Eindhoven, The Netherlands. [4]Laboratory of Physical Chemistry, Department of Chemical Engineering & Chemistry, Center for Multiscale Electron Microscopy and Institute for Complex Molecular Systems, Eindhoven University of Technology, 5600 MB Eindhoven, The Netherlands. [5]College of Polymer Science and Engineering, Sichuan University, Chengdu 610065, PR China. [6]These authors contributed equally: Jianhong Wang, Hanglong Wu. ✉e-mail: J.Shao@tue.nl; J.C.M.v.Hest@tue.nl

level of spatial and temporal control thermophoretically propelled nanomotors have been intensively studied[30–32]. In particular gold (Au)-based nanomotors have been investigated[33] and to date, a wide range of Au morphologies have been introduced into nanomotor systems, including spheres[34,35], shells[36,37], rods[38,39], and stars[40,41] via covalent conjugation or sputter coating techniques. However, the current existing nanomotor systems still hold some technical challenges. First of all, the deposition of the gold component on the nanomotor chassis is difficult to control spatially[42]. Secondly, most Au-based structures are larger than 10 nm, whereas a decrease in size could be highly beneficial for the efficiency of energy conversion and for many applications in nanomedicine[43–45]. Furthermore, smaller sized particles would allow for a more controlled deposition of these active components on the nanomotor chassis[46,47].

In recent years, we have developed a nanomotor platform based on biodegradable bowl-shaped polymersomes, or stomatocytes, which are composed of poly(ethylene glycol)-b-poly (D, L-lactide) (PEG-PDLLA) building blocks[8,48,49]. Block copolymer self-assembly and subsequent dialysis-mediated shape transformation provided us with an inherent asymmetric morphology, which guarantees directional movement when propelled by external stimuli. Additionally, the streamlined shape of the stomatocytes could also be beneficial for their movement[50]. To construct nanomotors, we have equipped these stomatocytes for example with manganese dioxide particles in the cavity[51], or with a gold hemispherical shell on the stomatocyte surface[37].

Herein we describe a gold-functionalized stomatocyte nanomotor design with unsurpassed motility. This was achieved by optimally benefiting from the photothermal features of Au nanoparticles, by decorating the outer surface of the stomatocytes with a tightly packed layer of ~ 5 nm Au NPs (Fig. 1). By analyzing their motion, we found that light propelled Au-stomatocyte nanomotors displayed excellent motility in both pure water and biological medium (PBS and DMEM), with maximum velocities of $124.7 \pm 6.6 \, \mu m \, s^{-1}$, $109 \pm 3.3 \, \mu m \, s^{-1}$, and $104 \pm 3.7 \, \mu m \, s^{-1}$, respectively. To the best of our knowledge, our light-activated stomatocyte-based nanomotors thereby outperform currently reported nanomotor systems. In order to explain this feature, the 3D morphology of the nanomotors was analyzed in detail by quantitative cryo-electron tomography (cryo-ET), which revealed that the bowl-shaped stomatocytes were endowed with an asymmetric arrangement of Au NPs along the z-axial direction, resulting in uneven distribution of plasmonic heating upon laser irradiation. Consequently, the Au-stomatocytes exhibited directional motion with high velocities. To demonstrate their potential in biomedical application, Au-stomatocytes based nanomotors were used for intracellular delivery of cargoes, including FITC-BSA and Cy5-siRNA into the cytoplasm by directly crossing the cell membrane as a result of their mechanical disruption.

## Results and Discussion

To construct the nanomotor chassis, three biodegradable PEG-PDLLA block polymers ($PEG_{22}$-$PDLLA_{95}$, $PEG_{44}$-$PDLLA_{95}$, and $NH_2$-$PEG_{67}$-$PDLLA_{95}$) were selected as building blocks and synthesized via our previously reported ring-opening polymerization method[51]. The composition and polydispersity index (PDI) of the block copolymers were characterized via proton nuclear magnetic resonance spectroscopy ($^1H$ NMR) and gel permeation chromatography (GPC) (Supplementary Fig. 1 and Supplementary Table 2). Thereafter, polymersomes were first prepared by self-assembly of the copolymers (weight ratio is 5:4:1, respectively) via the solvent switch methodology[52]. Bowl-shaped stomatocytes were then obtained by osmotic-induced shape transformation via dialysis of polymersomes against 50 mM NaCl solution. The two non-functional block copolymers were chosen, as we previously found out that this combination allows for a facile shape change process. The amine-terminated PEG-PDLLA block polymer was introduced for the efficient decoration of Au NPs on the surface of the stomatocytes. The Au-NPs were prepared in-situ and were deposited on the outer surface of the stomatocytes via electrostatic and hydrogen bond interactions (Fig. 2a)[53]. The size and morphology of the stomatocytes were characterized using dynamic light scattering (DLS) and cryo-TEM (Fig. 2b, c and Supplementary Fig 2a), where the typical opening/neck after shape transformation was clearly observed. Cryo-TEM images (Fig. 2b and Supplementary Fig. 2b) furthermore showed that the outer stomatocyte membrane was fully covered by Au NPs. The surface charge of the stomatocytes before and after Au NPs deposition was measured and changed from $1.6 \pm 0.8$ to $-16.1 \pm 1.1 \, mV$, indicating that the Au NPs were effectively deposited (Supplementary Table 3). Furthermore, the color of the sample solutions changed from white cloudy (stomatocytes) into a deep-red solution (Au-stomatocytes) after Au NPs deposition (Supplementary Fig. 3). Owing to the presence of Au NPs, the hydrodynamic size and PDI of the Au-stomatocytes were larger than the stomatocytes (Fig. 2c). UV-vis extinction spectroscopy of the Au-stomatocytes indicated the maximum absorption of the Au-stomatocytes was ca. 540 nm (Fig. 2d). As a control group, Au NPs deposited on spherical polymersomes (Au-polymersomes) were also prepared by using the same method as for the Au NP decorated

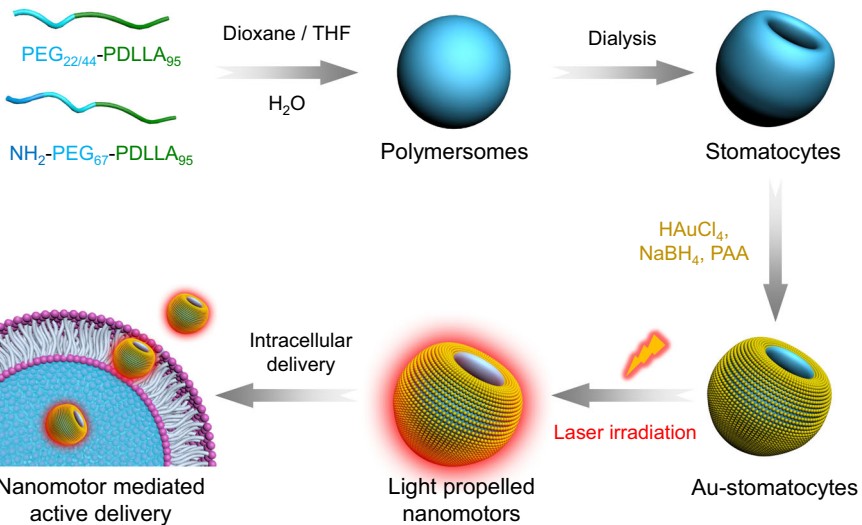

**Fig. 1** | Schematic illustration of the preparation of light-propelled biodegradable stomatocyte nanomotors for efficient intracellular transport.

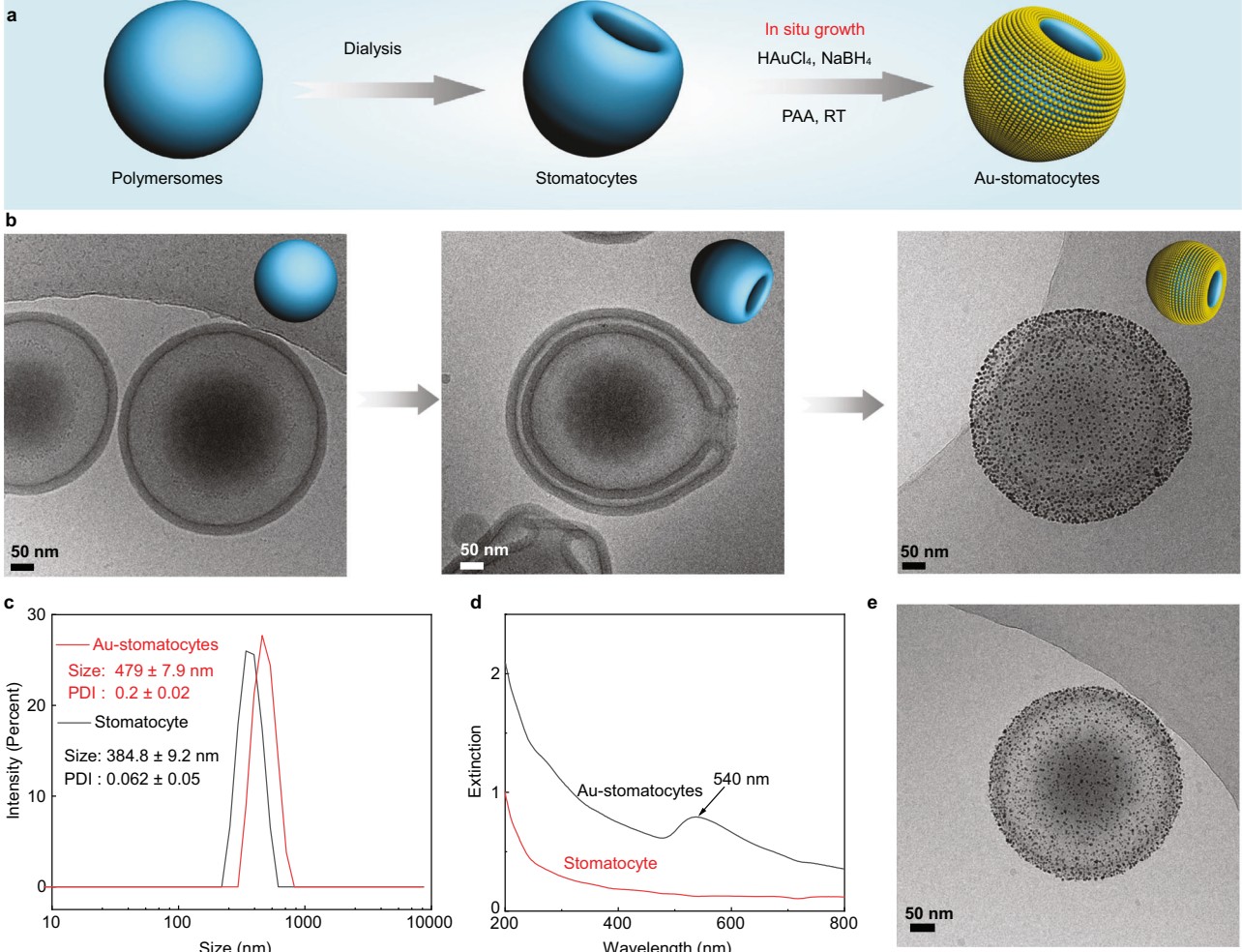

**Fig. 2 | Preparation and characterization of stomatocytes and Au NPs functionalized stomatocytes (Au-stomatocytes). a** Design strategy of Au-stomatocytes. **b** Cryo-TEM analysis of spherical polymersomes (left) and bowl-shaped stomatocytes (middle) and Au NPs coated stomatocytes (right), 5 experiments were repeated independently with similar results. **c)** Average size and PDI of stomatocytes and Au-stomatocytes measured by DLS. **d** UV-vis extinction spectra of stomatocytes (red line) and Au-stomatocytes (black line). **e** Cryo-TEM image of a typical spherical Au-polymersome, 5 experiments were repeated independently with similar results. All scale bar = 50 nm.

stomatocytes and were characterized by DLS, UV-vis spectra, and cryo-TEM (Fig. 2e and Supplementary Fig. 4).

To evaluate the photothermal performance of Au-stomatocytes upon laser irradiation, they were irradiated with a 660 nm laser. As a control, unmodified stomatocytes were treated under the same conditions. The temperature of an aqueous dispersion of Au-stomatocytes upon laser irradiation (output laser power = 1.5 W, see the corresponding laser power density in Supplementary Table 4) increased strongly and depended on the sample concentration (Fig. 3a) with the maximum temperature change (ΔT) of Au-stomatocytes (3.33 mg mL⁻¹) being 27.2 K within 10 min. The apparent temperature increase clearly demonstrated that the Au-stomatocytes displayed an efficient photothermal effect. In contrast, the temperature change of pure stomatocytes (ΔT = 10.5 K) and pure water (ΔT = 6.8 K) were significantly lower (Fig. 3b). As expected, the plasmonic heating was found to be related to the output laser power and laser exposure time (Fig. 3c), indicating that the photothermal effect could be readily regulated. These results demonstrated that Au-stomatocytes efficiently and rapidly converted absorbed laser energy into thermal energy. Next, we evaluated the photothermal stability of Au-stomatocytes via cyclic heating-cooling measurements. As shown in Fig. 3d, the temperature increase was stable after 5 cycles, which proved good photothermal stability of the Au-stomatocytes. The

photothermal heating of each groups (Au-stomatocytes, pure stomatocytes, pure water, and Au-polymersomes) upon laser irradiation was also visually observed with an IR camera, as shown in Fig. 3e and Supplementary Fig. 5. Furthermore, the structural stability of Au-stomatocytes after laser irradiation (1 W and 1.5 W) for 10 min was evaluated by cryo-TEM. As shown in Supplementary Fig. 6, the morphology of the Au-stomatocytes was also intact after this treatment. Consequently, we can conclude that Au-stomatocytes exhibited outstanding thermal adjustability and stability upon laser irradiation.

Next, we investigated the autonomous motion of Au-stomatocytes by irradiating the particles with a 660 nm laser, and following their motile behavior by nanoparticle tracking analysis (NTA) (See the experimental set-up in Supplementary Fig. 7). Based on the trajectories, the corresponding mean square displacement (MSD) and velocity were deduced using Golestanian's self-diffusiophoretic model[54]. Upon laser illumination, Au-stomatocytes exhibited directional autonomous motion which was opposite to the laser source (Fig. 4a). As reported previously, the possible mechanism of the observed negative phototaxis could be attributed to the asymmetric structure and non-homogenous plasmonic heating around the Au-stomatocytes[8,55]. Furthermore, compared to a series of control groups (300-nm Au NPs, pure stomatocytes, and Au-polymersomes), only Au-stomatocytes displayed directional autonomous motion with very

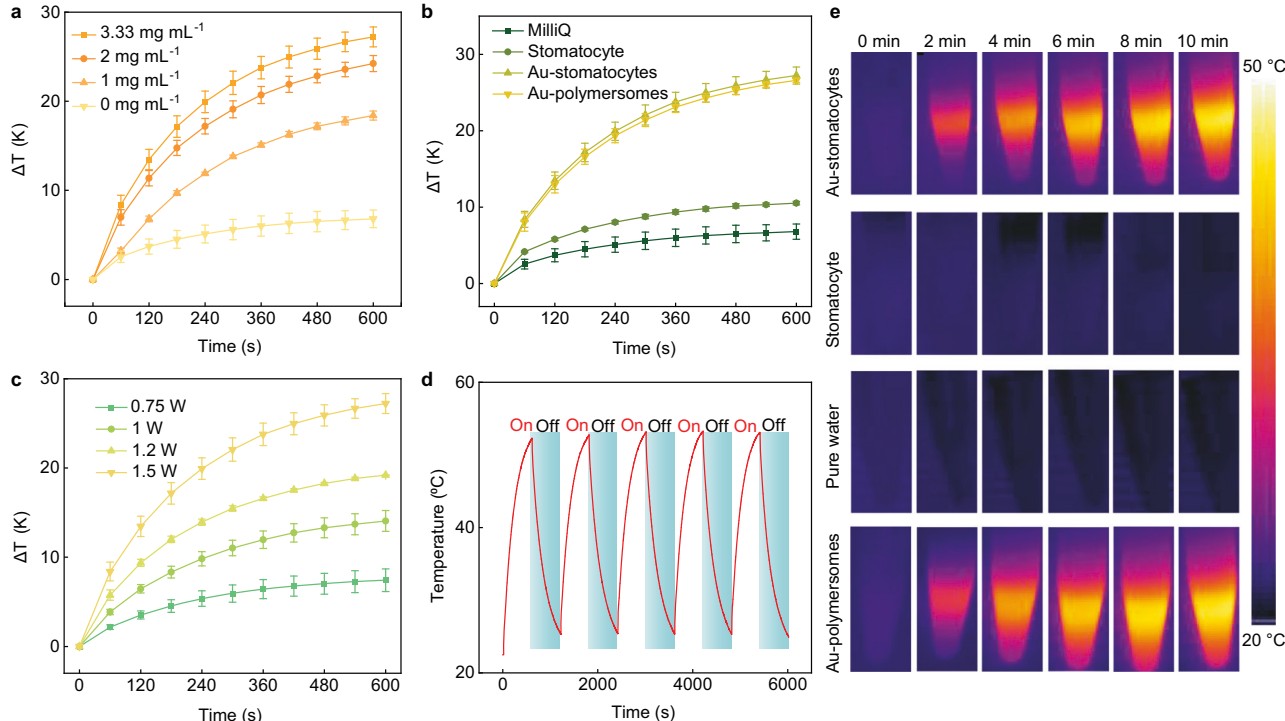

**Fig. 3 | Photothermal performance of Au-stomatocytes and control particles upon laser irradiation. a** Temperature changes of Au-stomatocytes at different particle concentrations (0 mg mL⁻¹, 1 mg mL⁻¹, 2 mg mL⁻¹, and 3.33 mg mL⁻¹) upon laser irradiation (660 nm, 1.5 W, 10 min), data represent mean ± SD for n = 3 independent samples. Error bar represents the standard deviation (n = 3). **b** Temperature change profiles of irradiated aqueous media (660 nm laser irradiation, 1.5 W, 10 min) containing pure water, stomatocytes, Au-stomatocytes, and Au-polymersomes (all at concentrations of 3.33 mg mL⁻¹), data represent mean ± SD for n = 3 independent samples. Error bar represents the standard deviation (n = 3).

**c** Temperature change of an aqueous sample containing Au-stomatocytes (3.33 mg mL⁻¹) as a function of input laser power (660 nm), data represent mean ± SD for n = 3 independent samples. Error bars represent the standard deviation (n = 3). **d** Photothermal stability of an aqueous sample containing Au-stomatocytes (3.33 mg mL⁻¹) upon cyclic laser irradiation (660 nm, 1.5 W). **e** Infrared thermographic maps of aqueous samples containing Au-stomatocytes, pure stomatocytes, pure water, and Au-polymersomes at a concentration of 3.33 mg mL⁻¹ as a function of irradiation time under 660 nm irradiation (1.5 W).

high velocities, whereas Au-polymersomes, 300 nm Au NPs, and pure stomatocytes only exhibited enhanced Brownian motion (Fig. 4b, c, Supplementary Movie 1 and 2). Furthermore, the motion behavior of Au-stomatocytes was highly dependent on the input laser power, as shown in Fig. 4d–f, Supplementary Table 5, and Supplementary Fig. 8 (Supplementary Movies 3 and 4). Au-stomatocytes displayed controllable motion as a function of laser power, and an ultrafast speed of 124.7 ± 6.6 μm s⁻¹ could be achieved with an output laser power of 1.5 W. Besides velocity, directionality of the movement of Au-stomatocytes could be manipulated as well by changing the incident laser pathway (Fig. 4g and Supplementary Movie 5). Notably, the motility of the Au-stomatocyte nanomotors was robust as no significant changes in MSD could be observed during 5 cyclic laser irradiations (Fig. 4h and Supplementary Fig. 9). As we were interested in potential biomedical applications, we further investigated the movement of Au-stomatocytes in relevant biological media (PBS and DMEM). Our results showed that they moved slightly faster in pure water compared to PBS and DMEM, as shown in Fig. 4i, which is consistent with previous findings[56]. The difference in speed was however relatively small, showing that the motile properties of Au-stomatocytes were retained in a biological context.

The remarkably high velocity of our Au-stomatocytes prompted us to investigate in more depth regarding the cause for this behavior. As gold-coated spherical polymersomes showed a similar heating profile as the stomatocytes, the reason had to be sought in the distribution of Au nanoparticles (Au NPs) over the stomatocyte surface, which should give rise to a well-defined temperature gradient[57]. Consequently, we firstly systematically measured the size of the Au NPs on

the outer surface of Au-stomatocytes via cryo-TEM. Approximately 500 Au NPs were analyzed, with an average diameter of 4.5 ± 1.2 nm (Fig. 5a, b). This value was corroborated by high-resolution dry-TEM of Au NPs that were isolated from the stomatocytes, by removing the block copolymers using THF (4.64 ± 1.54 nm, Supplementary Fig. 10).

To understand the spatial distribution of Au NPs at single stomatocyte level, cryo-ET[58,59] was carried out on a typical Au-stomatocyte (Fig. 5c–m, Supplementary Figs. 11, 12, and Supplementary Movie 6). Our cryo-ET data revealed that (1) the whole stomatocyte was fully covered with Au NPs (Fig. 5c–h); (2) the opening, or the neck of the stomatocyte was not completely filled with Au NPs, while only a few Au NPs were found in the narrowest part of the neck (Fig. 5c, e, h). (3) Only a few tens of Au NPs were found inside the cavity (Fig. 5h). To further quantify the 3D volume distribution of Au NPs, we performed quantitative analysis on the cryo-ET data, through which a 3D position and volume map were created (Fig. 5i and Supplementary Fig. 12). By summing the volume of the Au NPs and assuming that the standard volume of a single Au NP is ~ 50 nm³, the amount of Au NPs on the outer surface of this stomatocyte could be estimated to be ~6000, while only 20 Au NPs were in the cavity and ~ 30 Au NP inside the neck region (Fig. 5j). Furthermore, cross-sections from two different Z-heights in the 3D volume map (Fig. 5k) suggest that the Au NPs were homogeneously distributed radially in the X-Y plane, which was further supported by the radial average maps (Supplementary Fig. 13) shown in Fig. 5l. In contrast, regarding the Au distribution along the Z-axis, our quantitative analysis indicates that the Au density at the bottom of the stomatocyte (~1.9 × 10⁴ N μm⁻²) is higher than that at the opening

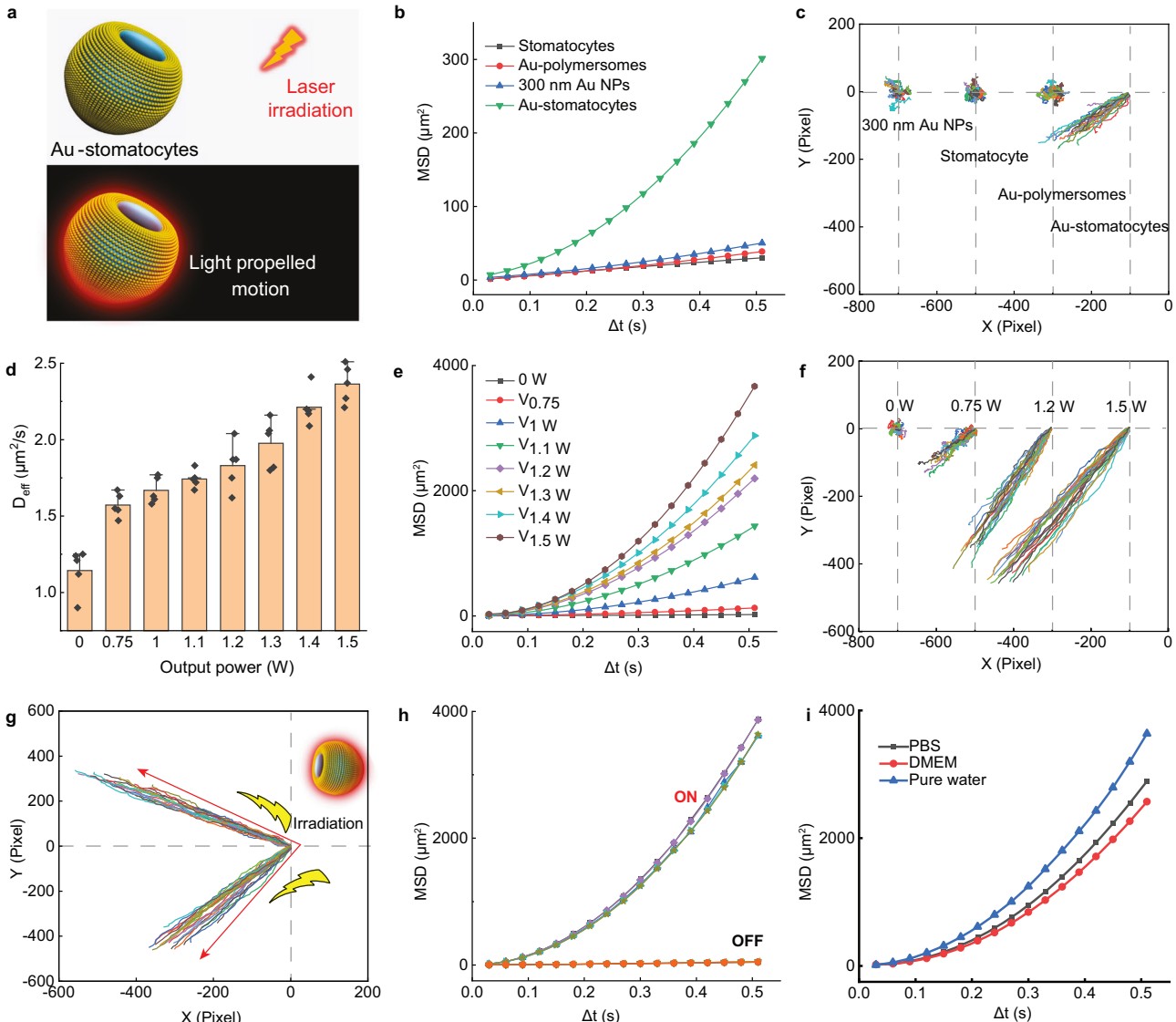

**Fig. 4 | Motion characterization of Au-stomatocytes. a** Schematic illustration of light triggered motility of Au-stomatocytes. **b** Mean square displacement (MSD) of Au-stomatocytes, and control groups (pure stomatocytes, 300 nm Au NPs, and Au-polymersomes). **c** Corresponding motion trajectories of different samples upon laser irradiation (1 W). **d** Diffusion coefficients of Au-stomatocytes as a function of output laser power, data represent mean ± SD for $n = 3$ independent samples, Error bars represent the standard deviation ($n = 3$). **e** MSDs of Au-stomatocytes as a function of output laser power. **f** Trajectories of Au-stomatocytes upon laser irradiation with different output laser power. **g** Controllable movement direction of Au-stomatocytes under laser illumination (660 nm, 1.5 W). **h** MSD of Au-stomatocytes with cyclic motion during five cycles of on-off laser irradiation. **i** MSD of Au-stomatocytes in different media (pure water, PBS, DMEM) upon laser irradiation (660 nm, 1.5 W).

($-8.5 \times 10^3$ N µm$^{-2}$) (Supplementary Figs. 14, 15). We speculate that during Au deposition, the membrane in the neck area has limited access to the Au precursor due to the confined space inside the cavity compared to the membrane outside the stomatocyte, resulting in a lower formation of Au NPs in the neck area. This hypothesis is supported by the fact that only tens of Au NPs are present in the cavity and in the narrowest part of the neck. This finding strongly indicates that upon NIR irradiation, the temperature at the bottom of the stomatocyte should be higher because of the higher local NP density, and more importantly, there should be a well-defined temperature gradient along the axial direction (the z-axis), which could explain the remarkable motile properties.

To understand the temperature distribution around a moving Au-stomatocyte at various velocities, we conducted a finite element method (FEM) simulation using the Au density distribution data from the cryo-ET results (Supplementary Fig. 16). Our simulation results confirmed the existence of a temperature gradient ∇T along the axial direction of the stomatocyte. For example, the average temperature gradient ∇T with an output power of 1.5 W was calculated to be ~100 µK µm$^{-1}$ (Fig. 5n and Supplementary Table 6). By calculating the drag forces exerted on the stomatocyte at different speeds, we found that overcoming drag forces at a speed of 125 µm s$^{-1}$ only requires a propulsion force of ~0.3 pN. Importantly, our analysis also revealed a linear relationship between the temperature gradient ∇T and the laser power input, and the calculated drag force ($F_d$) was also found to be almost linear with the temperature gradient (Supplementary Table 6 and Supplementary Fig. 17). This correlation can be mathematically expressed as $F_d = C\nabla T$, where C represents a constant coefficient. This finding strongly suggests that the propulsion mechanism of the Au-stomatocyte is primarily driven by the thermophoresis effect.

Due to their excellent motility in physiologically relevant media, Au-stomatocytes could be used to deliver cargos efficiently to cells via

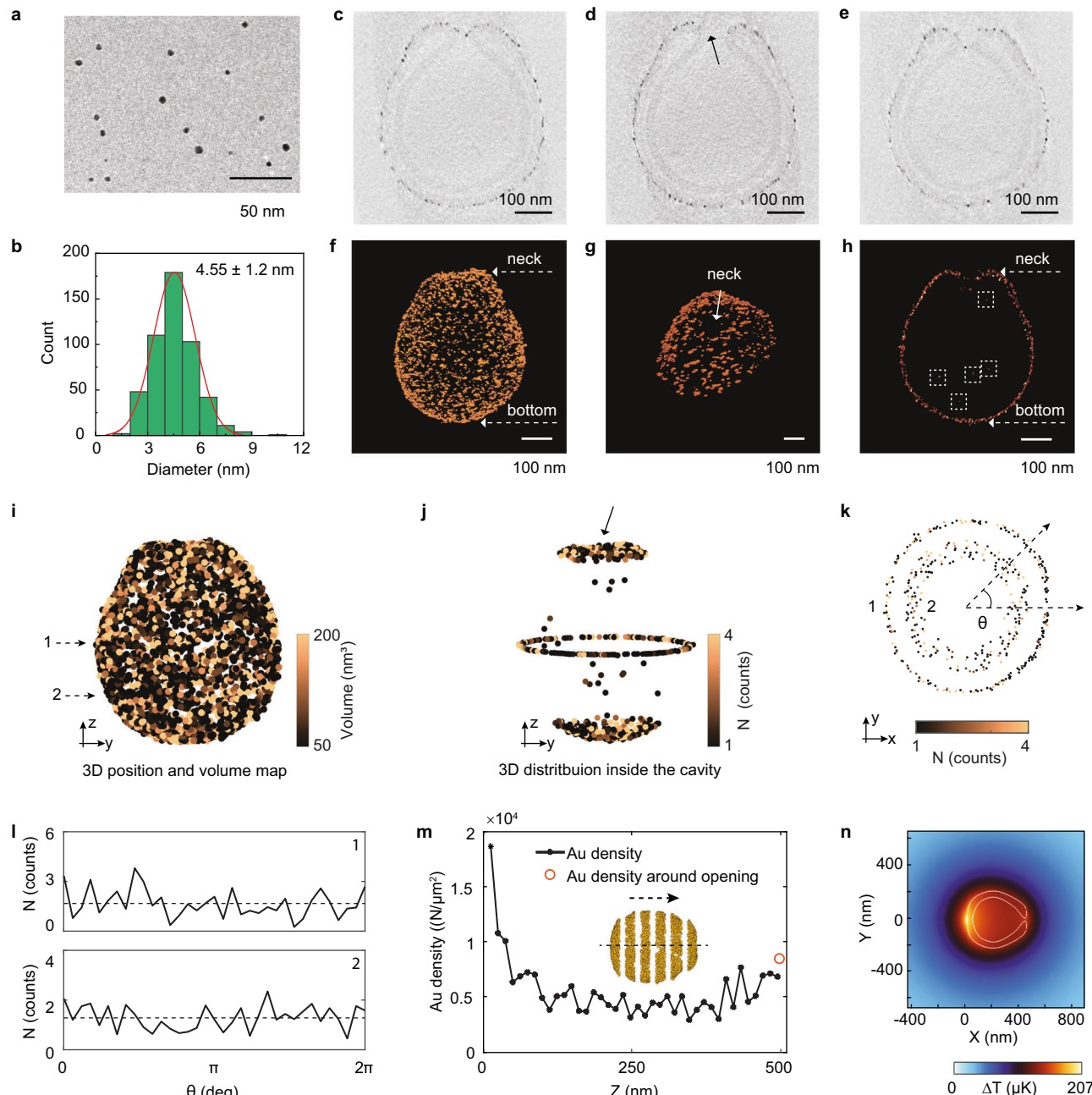

**Fig. 5 | Analysis of the size and spatial distribution of Au NPs by cryo-TEM and cryo-ET. a** Cryo-TEM image of as-prepared Au NPs (from the free-growth sample), 3 experiments were repeated independently with similar results, scale bar = 50 nm. **b** Size distribution of Au NPs coated on the stomatocytes. 500 Au NPs were measured. **c–e** Cross-sections of a Au-stomatocyte from cryo-ET. The arrow in d shows a Au NP trapped in the opening of the stomatocyte, 3 experiments were repeated independently with similar results, scale bar = 100 nm. **f–h** Volume rendering of a single Au-stomatocyte: (**f**) Overview; (**g**) Neck; (**h**) Cross-section showing the Au NP distribution in the neck and inside the cavity of an Au-stomatocyte (indicated with the dashed boxes), scale bar = 100 nm. **i–m** Quantitative analysis of the 3D Au NP distribution in a Au-stomatocyte. **i** 3D position and volume map of Au NPs on the outer surface of a Au-stomatocytes. **j** 3D distribution of Au NPs inside the cavity of a Au-stomatocyte. **k** 2D Au distribution from two different positions indicated by the dashed arrows. **l** Angular maps showing the Au NP distribution from two different cross-sections depicted in (**k**). **m** Changes in Au density along the z-axis. Inset: schematic showing how a stomatocyte is divided into several segments to estimate the Au density in each segment (see details in the Supplementary Information). The arrow indicates the moving direction of the stomatocyte under laser irradiation. **n** Simulated temperature distribution around a single Au stomatocyte based on the cryo-ET results (see details in the Supplementary Information).

direct translocation across the cell membrane. Before evaluating their performance in intracellular delivery, we first assessed the cytotoxicity of Au-stomatocytes using a CCK-8 assay. As shown in Supplementary Fig. 18, three cell lines including two cancer cell lines (HeLa and 4T1) and one healthy cell line (NIH/3T3) were tested. Au-stomatocytes exhibited good biocompatibility and the cell viability was high (90%) among all the cells and concentrations measured. Next, we

systematically investigated the intracellular delivery of Au-stomatocytes using HeLa cells. To further establish which laser power could be used without cell damage, we firstly investigated the cell viability when cells were incubated with different Au-stomatocyte concentrations upon laser irradiation (660 nm and 808 nm, 1 W, 5 min). Under this condition, all the cell viability was up to 80% (Supplementary Fig. 19). By encapsulating doxorubicin (DOX) as model

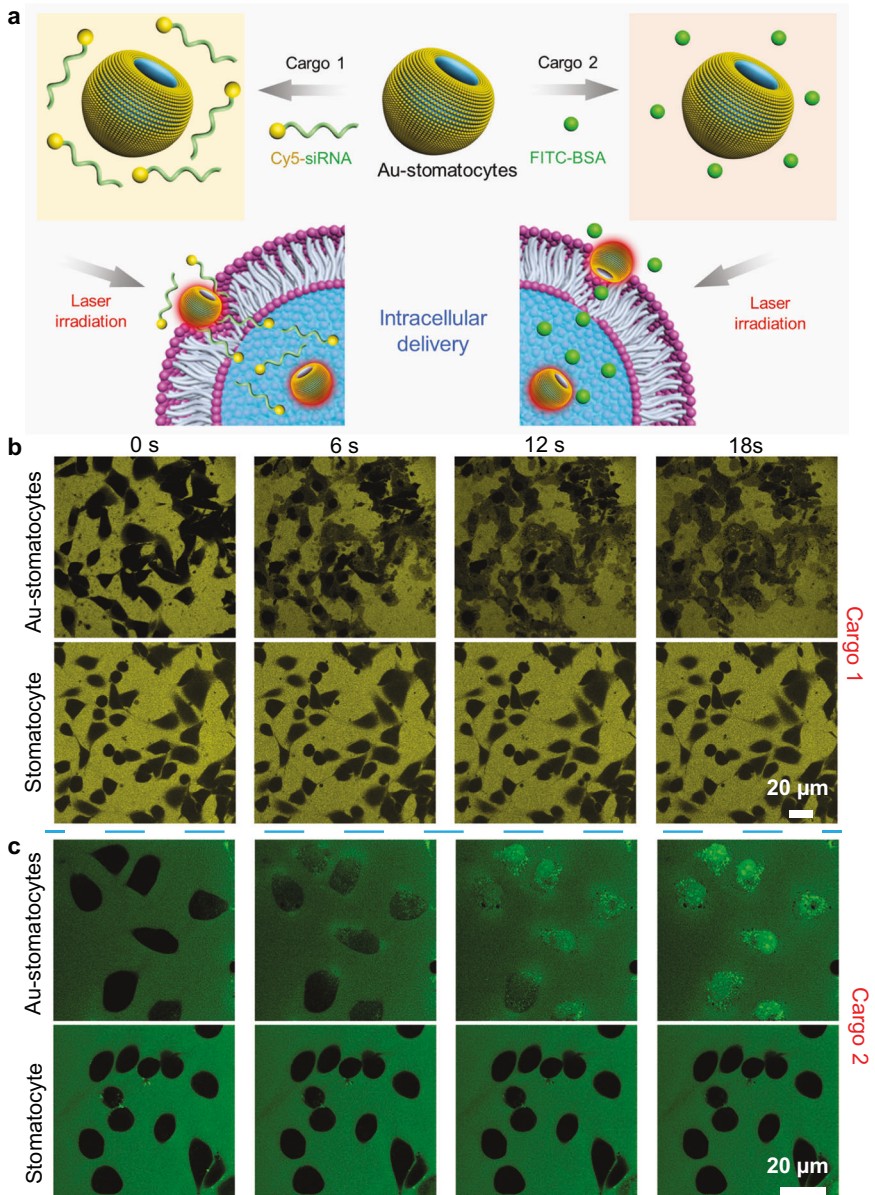

**Fig. 6 | Intracellular delivery of Cy5-siRNA and FITC-BSA mediated by Au-stomatocyte nanomotors upon 800 nm laser irradiation. a** Schematic illustration of light propelled Au-stomatocytes for intracellular delivery of siRNA and FITC-BSA, respectively. **b** Time-lapsed CLSM images of the transport of Cy5-siRNA into HeLa cells via Au-stomatocytes and pure stomatocytes upon laser irradiation. Yellow fluorescence originates from Cy5-siRNA, 3 experiments were repeated independently with similar results. **c** Representative CLSM images of delivering FITC-BSA to HeLa cells in the presence of Au-stomatocytes and pure stomatocytes under laser irradiation. Green fluorescence represents FITC-BSA, 3 experiments were repeated independently with similar results. All the scale bars = 20 μm.

hydrophobic therapeutic agent, it was possible to demonstrate the loading capacity of the Au-stomatocytes and to track their uptake in HeLa cells using fluorescence. DOX loading didn't affect the morphology of the stomatocytes, as confirmed by cryo-TEM (Supplementary Fig. 20). Furthermore, due to the fluorescence of DOX, we could directly visualize the autonomous motion of Au-stomatocytes using two photon confocal laser scanning microscopy (TP-CLSM) (Supplementary Movie 7 and 8). To study the interaction of Au-stomatocytes with HeLa cells, the cell membrane was first stained using a cell membrane marker (wheat germ agglutinin, Alexa Fluor™ 488 conjugate, WGA-AF488). After refreshing the cell culture medium with living cell imaging solution, Au-stomatocytes were introduced into the medium and the interaction between Au-stomatocytes and HeLa cells upon laser irradiation (800 nm) was immediately characterized using TP-CLSM. Time-lapse CLSM images in Supplementary

Fig. 21 demonstrated that Au-stomatocytes rapidly accumulated in the cell, as indicated by the increase of the fluorescence intensity over time.

As the direct translocation of the Au-stomatocyte nanomotors into the cell is accompanied by a temporary disruption of the cell membrane, it was investigated whether this could be used to improve the cellular uptake of two biomolecules, which are normally not capable of effective cell penetration. We chose fluorescein isothiocyanate labeled Bovine Serum Albumin (FITC-BSA, 66 kDa) and Cyanine 5 labeled small interfering RNA (Cy5-siRNA, Mw = 17852.4 Da) for this purpose (Fig. 6a), which are not taken up because of size and charge, respectively. siRNA is a class of therapeutic agents with much potential in biomedicine. However, it remains notoriously challenging to deliver these nucleic acids in the cytoplasm, which hampers their applicability. Uptake of the fluorescently labeled biomolecules was followed with

TP-CLSM. Au-stomatocytes and Cy5-siRNA or FITC-BSA were added to the HeLa cells, which were immediately irradiated with a 800 nm laser. Time-lapsed CLSM images in Fig. 6b, c showed the rapidly increasing fluorescence in HeLa cells induced by the delivery of respectively Cy5-siRNA and FITC-BSA in the presence of Au-stomatocytes upon laser irradiation. Within a short time frame (6 s), Cy5-siRNA was already transported into HeLa cells. In the control experiments no cellular uptake was observed (Supplementary Fig. 22). Consequently, the as-prepared Au-stomatocyte nanomotors can mediate the delivery of a wide range of compounds with varying size and charge.

Having proven that Au-stomatocytes can actively transport cargoes of different scales, we then studied the accumulation and penetration behavior of Au-stomatocytes in 2D cell cultures and a 3D HeLa spheroid model. Previously prepared DOX-loaded Au-stomatocytes were used to monitor their behavior in the in vitro environment. The HeLa cells were treated with DOX loaded Au-stomatocyes with varying output laser power (0, 1, and 1.5 W) for 5 min. Subsequently, the cells were imaged by CLSM after being co-cultured for an additional 6 h. As expected, an increase in red signal was observed with an increasing output laser power (Supplementary Fig. 23). Furthermore, 3D HeLa spheroids were cultured to investigate the capability of our Au-stomatocytes to penetrate biological barriers. Confocal z-scanning imaging sequences were carried out to study the fluorescence distribution and intensity in 3D spheroids. As representative results at depths of 80 and 100 μm, enhanced accumulation and penetration were observed with increasing output laser (Supplementary Fig. 24). These results indicate that the accumulation and penetration of tumor tissues are strongly associated with the nanomotors' velocity. Hence, developing an ultrafast nanomotor will provide a new approach to achieving enhanced deep tissue penetration and effective improvement in the accumulation of delivery vehicles in tumors.

In summary, we have designed a light-propelled biodegradable nanomotor (Au-stomatocytes) by decorating the surface of bowl-shaped stomatocytes with of Au NPs via electrostatic and hydrogen bond interactions. Upon laser irradiation, these nanomotors displayed excellent photothermal properties owing to the gold induced plasmonic heating and reached high velocities (125 μm s$^{-1}$) upon laser illumination with an output power of 1.5 W. Through quantitative analysis of the spatial distribution of the Au NPs on the Au-stomatocytes via cryo-TEM and cryo-ET, we could elucidate that the origin of these motile features should be the uneven distribution of Au NPs along the axial direction, which can generate a well-defined temperature gradient, resulting in a sub-pN force capable of propelling the nanomotor at such a high speed. The motile features of the nanomotors were furthermore exploited to achieve translocation of the particles in mammalian cells via the temporary disruption of the cell membrane. When this process was performed in presence of biomolecules, these were also effectively taken up, without affecting cell viability.

## Methods
### Instruments
The copolymers were analyzed with a Bruker AV 400 MHz Ultra-shieldTM spectrometer and Prominence-I GPC system (Shimadzu) with a PL gel 5 μm mixed D (Polymer Laboratories), equipped with a RID-20A differential refractive index detector. The hydrodynamic size and dispersity index (PDI) of the nanoparticles were determined by a Malvern instruments Zetasizer (model Nano ZSP) dynamic light scattering (DLS) equipped with a 633 nm He-Ne laser and avalanche photodiode detector. The morphologies of the formed nanoparticles were recorded with a FEI Quanta 200 3D FEG scanning electron microscopy (SEM). Cryogenic transmission electron microscopy (cryo-TEM) and cryo-electron tomography (cryo-ET) experiments were conducted on the TU/e CryoTITAN (Thermo Fisher Scientific) equipped with a field-emission gun operating at 300 kV, an autoloader station and a post-

column Gatan energy filter. Fluorescent images were recorded using a Confocal Laser Scanning Microscopy (CLSM, Leica TCS SP8X) equipped with two-photon laser source (Chameleon Vision, Coherent, USA). Cell viability was evaluated via a microplate reader (Safire2, TECAN). Nanosight Tracking Analysis was performed on a Nanosight NS300 equipped with a laser channel (488 nm) and sCMOS camera.

### Synthesis of block copolymers
The PEG-PDLLA copolymers were synthesized via ring opening polymerization (ROP), according to a previous reported protocol[52] with a slight change (Supplementary Fig. 25). Briefly, the macroinitiator monomethoxy-poly(ethylene glycol)-OH (1 K and 2 K) and D,L-Lactide (DLL) (PEG$_{22}$-PDLLA$_{95}$: 200 mg PEG 1 K, 2.736 g DLL; PEG$_{44}$-PDLLA$_{95}$: 200 mg PEG 2 K, 1.368 g DLL) were weighted into a round-bottom flask (100 mL) equipped with a stirring bar. Then, the compounds were dried by dissolution into dry toluene (50 mL, twice) followed by evaporation before polymerization. Thereafter, the dried compounds were co-dissolved in dry dichloromethane (DCM) ([monomer] = 0.5 M) under argon. Next, 1,8-diazabicyclo[5.4.0]undec-7-ene (DBU) (0.5 equiv. to [initiator]; 0.1 mmol = 15 μL) was added to the reaction mixture. The reaction was stirred for 2–4 h at room temperature (RT) under argon. The reaction progress was monitored by $^1$H NMR spectroscopy until the monomer peaks had disappeared. After finishing the polymerization, the reaction mixture was diluted with DCM and extracted with KHSO$_4$ (1 M, 2 x) and brine (1 M, 1 x). The organic layer (lower layer) was collected and dried with Na$_2$SO$_4$, followed by filtering and concentrating. Then precipitation of the concentrated solution was performed in ice cold diethyl ether (100 mL) and the precipitate was collected via centrifugation (1500 g, 7 min) and lyophilized from 1,4 dioxane (10 mL) to yield a white powder (yield = 80–85%). The polymerization was analyzed by $^1$H NMR and GPC. $^1$H NMR (Chloroform-d): 5.15-5.3 ppm (-C = OCHCH$_3$-, 2H, multiplet), 3.63−3.7 ppm (-CH$_2$CH$_2$O-, 4H PEG, multiplet), 3.35-3.40 ppm (CH$_3$-PEG, singlet) and 1.55−1.65 ppm (C = OCHCH$_3$-, 6H, multiplet).

### Synthesis of NH$_2$-PEG$_{67}$-PDLLA$_{95}$
The synthesis of amino terminated PEG-PDLLA block polymer was carried out according to a previously published procedure (Supplementary Fig. 26)[48]. Briefly, Boc-NH-PEG-OH (3 K, 240 mg) was weighted into a dried round-bottom flask (100 mL) with D,L-Lactide (1.096 g), equipped with a stir bar. The compounds were then dried by dissolution in dry toluene, followed by solvent evaporation. Dry DCM (16 mL) and DBU (6 μL) were subsequently added under argon atmosphere. The reaction was kept stirring for 2-4 hours at room temperature. Disappearance of the monomer peaks was monitored via $^1$H NMR spectroscopy. After finishing the polymerization, the reaction mixture was diluted with excess DCM and the polymer extracted with KHSO$_4$ (1 M, 2x) and brine (1 M, 1x). The organic layer (lower layer) was collected and dried with Na$_2$SO$_4$, filtered and concentrated. To cleave the Boc-group, the concentrated oil was dissolved into 4 mL HCl in dioxane (4 M) and was left stirring for 2 hours at RT. The mixture was then evaporated to remove HCl, and the crude product was lyophilized from 1,4 dioxane (10 mL) to yield a white powder (yield = 75%). Disappearance of the protecting group Boc peak (1.41 ppm, 9H, singlet) was monitored by $^1$H NMR spectroscopy.

### Preparation of stomatocytes and polymersomes
In a 20 mL vial, NH$_2$-PEG$_{67}$-PDLLA$_{95}$, PEG$_{22}$-PDLLA$_{95}$ and PEG$_{44}$-PDLLA$_{95}$ copolymers (1:5:4 w/w/w%, 20 mg) were weighed and dissolved in 2 mL of a mixture of THF and dioxane (1:4 v/v). Thereafter, a stirring bar was added, and the vial was sealed with a rubber septum. Subsequently, the solution was stirred at 900 rpm for 30 min, then 2 mL of MilliQ water was added with a syringe pump (Chemyx Inc. Fusion 100 Syringe Pump) at a rate of 1 mL h$^{-1}$. Afterwards, the cloudy suspension was transferred into a pre-hydrated dialysis membrane (SpectraPor,

molecular weight cut-off: 12 –14 kDa, 2 mL cm$^{-1}$) and dialyzed at 4 °C against pre-cooled NaCl solution (50 mM, 2 L) for at least 24 h, with a NaCl solution change after 1 h to yield stomatocytes. For comparison, polymersomes were prepared by dialysis against MilliQ water for at least 24 h with a water change after 1 h at 4 °C. For DOX loading, 0.5 mg DOX was dissolved together with the block copolymers prior to self-assembly.

### General method for the preparation of Au-stomatocytes and Au-polymersomes

For the preparation of Au-stomatocytes, the gold nanoparticles were prepared by the in situ growth method and coated on the surface of stomatocytes by electrostatic interaction. Briefly, in a 20 mL vial, poly(acrylic acid) (PAA, 0.8 mg) and HAuCl$_4$.4H$_2$O (1.2 μL, 1 mg mL$^{-1}$) were added to 1 mL of stomatocyte solution (3.33 mg mL$^{-1}$). The solution was stirred at a rate of 200 rpm for 10 min at RT. Then, a solution of NaBH$_4$ (1 mL, 5 mM) was added dropwise and the mixture was stirred for another 10 min at RT. The solution was centrifuged down (4600 g, 10 min), supernatant was removed and MilliQ water was added. This procedure was repeated until the supernatant was clear. Finally, the precipitate was dispersed in 1 mL MilliQ water to obtain a stomatocyte concentration of 3.33 mg mL$^{-1}$. Samples were stored in the fridge prior to further use. Au-polymersomes as a control group were prepared according to the same procedure mentioned above.

### Au-stomatocytes-assisted intracellular transport

To evaluate the Au-stomatocytes mediated active intracellular delivery of biomolecules, pre-cultured cells (in μ-slide 8 wells) were randomly divided into three groups, which were subsequently treated with Au-stomatocytes, pure stomatocytes (without Au nanoparticles) and cells only, these under NIR irradiation. For both stomatocyte groups, 20 μL of a 3.33 mg mL$^{-1}$ nanoparticle solution was added separately to each well before TP-NIR irradiation. 10 μL Cy5-siRNA (9 μg μL$^{-1}$) and 50 μL FITC-BSA (10 μg μL$^{-1}$) were simultaneously added to all the samples, respectively. TP-CLSM (Leica TCS SP8X) was used to observe and record the active intracellular transportation of Cy5-siRNA and FITC-BSA.

### Penetration of DOX loaded Au-stomatocytes in 2D and 3D tumor models

The pre-seeded HeLa cells were mixed with DOX loaded Au-stomatocytes and then treated with an external 660 nm laser at different output laser power (0, 1 W, and 1.5 W) for 5 min, the cells were washed three time with PBS after being co-cultured for another 6 h. HeLa cells were stained with Hoechst 33342 for the observation of the cell nucleus. Then the confocal Z-stack scanning was carried out to image the cells.

3D HeLa cellular spheroids were used to further investigate the penetration capacity of DOX loaded Au-stomatocytes. 3D HeLa spheroids were constructed by a "hanging drop" technique according to a previously reported procedure[60]. Briefly, 600 μL of agarose solution (2%, w/v, PBS) was dropped into the mold of a 3D petri dish (Sigma-Aldrich). After 5 min, the solidified gels were carefully placed into each well of a 12-well plate from the mold and equilibrated for more than 20 min with DMEM. Then, a cell suspension (190 μL) containing 6 × 10$^5$ HeLa cells were slowly added to each well of the 12-well plate, and after standing for 10 min, 2.5 mL of fresh cell culture medium was slowly added to each well to let the cells aggregate and grow. When the tumor spheroids reached an appropriate volume, the penetration capacity of the Au-stomatocyte nanomotors was evaluated. The prepared 3D HeLa spheroids were mixed with DOX loaded Au-stomatocytes ([DOX] = 2 μg mL$^{-1}$), and following irradiation with an external 660 nm laser at different output laser (0, 1 W, and 1.5 W) for 5 min, the spheroids were washed three time with PBS after being co-cultured for an additional 6 h. Then the spheroids were imaged using confocal laser scanning microscopy (CLSM).

### Reporting summary

Further information on research design is available in the Nature Portfolio Reporting Summary linked to this article.

## Data availability

Additional data are available from the corresponding author upon request. Source data are provided with this paper.

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

## Acknowledgements

This study was financially supported by The Netherlands Ministry of Education, Culture and Science (Gravitation Program 024.001.035 to J.C.M.H.), the Spinoza premium to J.C.M.H., and the European Union's Horizon 2020 research to J.C.M.H. and innovation program Marie Sklodowska-Curie Innovative Training Networks (ITN) Nanomed (No. 676137) to J.C.M.H. J. Wang thanks the support from the China Scholarship Council. Author 1 and Author 2 contributed equally to this work.

## Author contributions

J.W. and H.W. contributed equally to this work. J.W., J.S, H.W., J.C.M.H. wrote the manuscript; J.W., J.S., H.W., and J.C.M.H. designed the research; J.W., J.S., H.W., Yingtong., R.R.M., S.C., H.F., Yudong. and

L.K.E.A.A. performed the experiments. H.W. developed and performed cryo-TEM and cryo-ET experiments and the quantitative analysis of the EM data. R.Z. and S.R.J.H. synthesized Cy5 modified siRNA. Yudong. developed the Python script for motion analysis. X.Z. and H.W. carried out the simulation of the heating map and all the related calculations. All the authors contributed to the critical discussion of the results and commented on the manuscript.

## Competing interests

The authors declare no competing interests.
