## [Peer Review File · Nature Communications]

REVIEWER COMMENTS

Reviewer #1 (Remarks to the Author):

The manuscript is written by an excellent group with an international reputation. The paper is well written, the figures look nice and the group's work is generally well cited. Unfortunately, I am shocked about the lack of attention to detail (the SI videos do not have indications what they show, the naming is not even S1, S2, ...) so there is no way to understand what is supposed to be what. Some of these videos clearly show flow, no active motion. Maybe it is a lack of diligence, but what is supposed to give the reader an impression of the actual research only makes me doubt.

The polymersomes have been known for more than a decade now, the internalization into cells has been published (refs 7-9) and the ultrafast motion seems to be an external flow.

Reviewer #2 (Remarks to the Author):

In this article, authors utilized the previous polymersomes stomatocytes and loaded with gold nanoparticles, achieving photothermal-driven nanomotors with a speed of 125 $\mu\text{m/s}$. They have achieved a breakthrough in the nanomotor's motion speed and explained it by the uneven loading of gold particles. However, the authors have not effectively presented the significance of the speed breakthrough, failed to provide a reasonable explanation for the uneven loading of gold particles, and have not explained why this uneven loading results in a significant speed increase. Considering lack of solid evidences to support the conclusions, we do not believe that it is currently suitable for publication in Nature Communications.

1. While it is true that the nanomotor's motion speed of 125 $\mu\text{m/s}$ in this article is significantly faster than the previously reported 80 $\mu\text{m/s}$, the authors need to present the significance of this speed with specific application scenarios. For instance, are there specific applications where nanomotor speeds of 100 $\mu\text{m/s}$ or higher are essential? In this article, the authors have merely verified that nanomotors can enter cells, with no exploration of the impact of speed on cell penetration.

2. The authors confirmed the uneven loading of gold particles on polymersomes stomatocytes through cryo-electron microscopy and slicing scans. However, the article stops here without explaining the relationship between this uneven loading and motion speed, especially the record-breaking speed.

3. Previous articles have reported uneven loading of gold or platinum nanoparticles on bottle-like nanomaterials, such as in Nano Lett. 2021, 21, 6071. However, they did not achieve such high motion speeds. Considering that most of these articles used silica particles, does this imply that the record-breaking speed in this article might be primarily attributed to the lightweight nature of polymersomes rather than the asymmetrical structure of gold nanoparticles?

4. Similarly, the authors have not provided an explanation for the causes of this uneven distribution of gold particles.

Reviewer #3 (Remarks to the Author):

Wang et al. report on the synthesis and dynamics of light activated nanomotors in their manuscript entitled "Ultrafast Light-activated Polymeric Nanomotors".

The authors thereby target the development of a novel active particle that is made from biodegradable polymers showing a bowl shape reminiscent of stomatocytes. The polymerosomes are decorated with gold nanoparticles of about 5 nm diameter, which makes them subject to photothermal heating. This optical induced heat generation is suggested to yield a very fast motion of these particles, which is determined to be around 120 $\mu\text{m/s}$. The study therefore belongs to the highly interesting field of active particles, which is currently intensely explored also with the aim of finding new smaller motors that can provide new functions.

The paper is well written explaining the main results on the dynamics of these particles and also extensions which include the invasion of these particles into cells. The synthesis and characterization by cryo-electron microscopy is impressive. However, the study of the dynamics of these active particles as well as their presentation reveals several issues:

1. The authors present data on the size of the particles, which is around 400-600 nm depending on if they are decorated with AuNP or not. It might be academic but I would count that as nanomotors. For me this is a particle with dimensions of less than 100 nm. Also, why is the size of the particles larger when decorated?
2. Particles of 500 nm diameter possess a rotational diffusion time of about 100 ms in water. This rotational diffusion time separates the short time limit of the mean squared displacement model used from the enhanced diffusive limit. In other words it separates the parabolic part of the MSD from the linear part. This is because the propulsion direction is typically connected to the geometry of the particle and rotates with the particle. Yet, there is no diffusive part visible from all the plots. Why is that? The absence of rotational diffusion points at additional forces that are relevant for the motion and raise doubts about the suggested propulsion mechanism. Also, why are the particles traveling under a specific angle to the light source?
3. Even though the authors report that the measurements of the particle dynamics were done with a NanoSight tracker, the overall geometry and observation is completely unclear. None would be able to reproduce the experiments except having a NanoSight tracker with exactly the same configuration then.
4. Why did the authors choose 660 nm for photothermal heating? 5 nm AuNP have commonly their plasmonic absorption maximum at about 520 nm.

5. The authors mention a power of 1.5 W that is used for the heating and propulsion experiments. Power is only useful when providing the illuminated area as well.
6. The authors report the absorption spectrum in Fig. 2d of the main text. Given the shape, I would rather think that the authors measured extinction, e.g., the removal of light from the incident direction, which also contains scattering.
7. Figure 3 seems to report the local average temperature of the sample as the IR images. Is the temperature observed there really corresponding to the temperature at the stomatocyte surface?

As the novel element is mainly the high velocity and the origin is unclear, I would not recommend this manuscript for publication in Nature Communication. As the synthesis and the study of structure is nice, I would suggest a more material oriented journal for publication.

Minor stuff:

1. The authors mention that the particles have a „shape of the stomatocytes leads to a lower drag coefficient that could be beneficial for their movement“. In fact active particles do not propel against friction and the shape is of other importance than for particles, which are dragged through the liquid.
2. The authors also mention that the particles cross the cell membrane due to their kinetic energy. Also this is not the correct physical picture.
3. Temperatures are commonly given in Kelvin or degree centigrade. Temperature difference are given in Kelvin.
4. The authors refer to Golestanian self-diffusiophoretic model. Actually this model is a general description of the mean squared displacement of a self-propelled Brownian particle and not bound to diffusiophoresis.
5. The authors report the surface charge of the stomatocytes before and after the AuNP deposition, but without giving a unit.
6. The axis labels in Figure 4c is unreadable and Figure 4 is in general of low quality with different font sizes everywhere. Also giving just X-position and Y-position as axis labels in Fig. 4G is not sufficient for a publication submitted to Nature Communications.

Reviewer #1

The manuscript is written by an excellent group with an international reputation. The paper is well written, the figures look nice and the group's work is generally well cited.

We sincerely appreciate this reviewer's nice words and positive comments on our work.

1. Unfortunately, I am shocked about the lack of attention to detail (the SI videos do not have indications what they show, the naming is not even S1, S2, ...), so there is no way to understand what is supposed to be what.

Captions for all the supplementary videos were provided in the Supplementary Information in our original version (**Page S19**). However, based on the reviewer's suggestion to enhance clarity for the readers, we have added corresponding indications for each SI video in the revised manuscript.

2. Some of these videos clearly show flow, no active motion. Maybe it is a lack of diligence, but what is supposed to give the reader an impression of the actual research only makes me doubt.

Flow is one of the main factors that indeed can affect motion. Therefore, to avoid the influence of flow, a sealed sample chamber has been used in our experiments. The set-up for observing the active motion of nanomotors via NanoSight is described in the **Methods Section (Page S4)**. To help readers better understand our experimental set-up, a scheme has been added to the Supplementary Information as **Scheme S3** (see below, also see **Page S10**). Furthermore, we have also designed a series of control groups to demonstrate that the nanomotors are propelled by light irradiation, instead of flow (**Fig. 4**). These control experiments include nanomotors with/without laser irradiation, pure stomatocytes (without gold coating) with laser irradiation, and pure 300 nm gold nanoparticles with laser irradiation. Our results from all these control experiments have strongly proven that the active motion of Au-stomatocytes is induced by asymmetric morphology and laser irradiation, rather than by flow.

Scheme S3. Schematic illustration of the experimental set-up for characterizing the motion of Au-stomatocytes using NanoSight equipped with an external 660 nm laser for irradiation.

Reviewer #2

In this article, authors utilized the previous polymersomes stomatocytes and loaded with gold nanoparticles, achieving photothermal-driven nanomotors with a speed of 125 $\mu\text{m/s}$. They have achieved a breakthrough in the nanomotor's motion speed and explained it by the uneven loading of gold nanoparticles. The authors have not effectively presented the significance of the speed breakthrough, failed to provide a reasonable explanation for the uneven loading of gold particles, and have not explained why this uneven loading results in a significant speed increase.

We thank the reviewer for the comments and agree that it is important to clarify the significance of the speed breakthrough, the cause of the uneven distribution of Au nanoparticles, and the relationship between the uneven Au loading and the ultrafast motion speed. Consequently, we have addressed each point individually in our response to comments 1, 3 and 4.

1. While it is true that the nanomotor's motion speed of 125 $\mu\text{m/s}$ in this particle is significantly faster than the previously reported 80 $\mu\text{m/s}$, the authors need to present the significance of this speed with specific application scenarios. For instance, are there specific applications where nanomotor speeds of 100 $\mu\text{m/s}$ or higher are essential? In this article, the authors have merely verified that nanomotors can enter cells, with no exploration of the impact of speed on cell penetration.

Nanomotors with an ultrafast velocity could lead to better tissue penetration, more efficient intracellular drug delivery, and enhanced accumulation in the targeted area. Several related references have been cited in the main text as Ref 27, 28, and 29 (*Sci. Robot.* **2022**, 7, eabo4160; *Appl. Mater. Today* **2023**, 34,101916; *Nat. Mater.* **2022**, 21, 1324-1332.). The importance of the speed breakthrough has been summarized in the **Introduction section (Page 3)**. Furthermore, as suggested by the reviewer, we performed additional in vitro experiments to verify that the Au-stomatocytes as nanomotors could enter cells upon laser irradiation. According to the results, we found that intracellular delivery could be enhanced via Au-stomatocytes with a higher motion speed. The corresponding description regarding the effect of motion speed on cell penetration has been added to the manuscript (**Page 20-21**), and the data as **Fig. S22** have been added to the Supplementary Information (**Page S18**). To further investigate the impact of speed on cell penetration, we conducted additional experiments to compare the penetration of nanomotors at different velocities (125 $\mu\text{m s}^{-1}$ and $<100 \mu\text{m s}^{-1}$) using 3D tumor spheres. Z-scanning CLSM images clearly demonstrated the deeper penetration of the nanomotors at faster speed (125 $\mu\text{m s}^{-1}$). Corresponding description for this part was added to the manuscript and related data was added as **Fig. S23** to the Supplementary Information (**Page S19**). All the experimental protocols were added to the Methods section in the manuscript (**Page 25-26**).

2. The authors confirmed the uneven loading of gold particles on polymersomes stomatocytes through cryo-electron microscopy and slicing scans. However, the article stops here without explaining the relationship between this uneven loading and motion speed, especially the record-breaking speed.

We appreciate the reviewer's comment and acknowledge the necessity of elucidating the correlation between uneven loading of gold nanoparticles and motion speed. The asymmetric distribution of gold nanoparticles along the axial direction of the stomatocyte should inherently introduce a temperature gradient in the same direction due to their photothermal effects. Consequently, we conducted a finite element method (FEM) simulation to comprehensively

understand the temperature distribution around a moving Au-stomatocyte at various velocities (**Fig. S15**, also see in the attached figure). Our simulation results confirm the existence of a temperature gradient along the axial direction of the stomatocyte, due to the uneven loading of the gold nanoparticles. The method of the FEM simulation was added as **Section 2.4** to the Supplementary Information (**Page S5-S6**).

To further understand the propulsion forces needed to attain high motion speeds, we calculated the drag forces exerted on the stomatocyte at different speeds, as the propulsion forces are equivalent to drag forces under low Reynolds number conditions. Our analysis reveals that overcoming drag forces at a speed of $125 \mu\text{m s}^{-1}$ only requires a propulsion force of $\sim 0.5 \text{ pN}$ (assuming the nanomotor is spherical) based on Stokes' law, or $\sim 0.3 \text{ pN}$ based on the solution of the full Navier–Stokes equations, considering the shape factor of the stomatocyte (**Table S6**). Importantly, our analysis reveals a linear relationship between the temperature gradient (∇T) and the laser power input, and the calculated drag force (F_d) is also found to be almost linear with the temperature gradient (**Table S6 and Fig. S16**, also see in the attached table and figure). This correlation can be mathematically expressed as $F_d = C\nabla T$, where C represents a constant coefficient. It is also worth noting that the steady-state temperature around the stomatocyte can be rapidly achieved within a few milliseconds which is consistent with our observation of instantaneous nanomotor motion when activated by the laser irradiation.

In conclusion, through FEM simulation and propulsion force analysis, we confirm that the uneven distribution of gold nanoparticles along the axial direction of the stomatocyte can generate a well-defined temperature gradient, resulting in a sub-pN force capable of propelling the nanomotors at ultrafast speed. A summary of the above discussion has been added to the main text (**Page 13-14**).

Fig. S15. Understanding the temperature distribution around a single Au-stomatocyte using Finite Element Method (FEM) simulations. **a**) Numerical configuration for 2D axisymmetric simulation with the corresponding boundary conditions. Plasmonic heating of NPs is modeled as a surface heat flux distributed on the stomatocyte's surface. z and r represent the axial and radial coordinates, respectively. U_s represents the stomatocyte's swimming velocity measured in experiments. Γ_{UW} and Γ_{DW} denote the upwind and downwind surfaces of the stomatocyte. **b**)

Simulated temperature field around a single Au-stomatocyte under 1.5 W laser irradiation. Note that here we show the temperature increase ΔT instead of the temperature.

Table S6 Overview of calculated surface averaged ∇T and drag forces at different laser output powers and velocities.

Output laser power (W)	Surface averaged ∇T ($\mu\text{K } \mu\text{m}^{-1}$)	Velocity ($\mu\text{m s}^{-1}$)	Drag force ($\times 10^{-14}$ N)
0.75	42.3	19.5	5.06
1.0	55.4	47.7	12.4
1.1	62.1	73.3	19.0
1.2	67.9	91.2	23.6
1.3	73.5	95.4	24.7
1.4	79.1	104.5	27.1
1.5	85.0	124.7	32.3

Fig. S16. a) The measured averaged temperature gradient ∇T around a moving stomatocyte as a function of laser output power. **b)** The calculated drag force exerted on the stomatocyte as a function of ∇T .

3. Previous articles have reported uneven loading of gold or platinum nanoparticles on bottle-like nanomaterials, such as in Nano Lett. 2021, 21, 6071. However, they did not achieve such high motion speeds. Considering that most of these articles used silica particles, does this imply that the record-breaking speed in this article might be primarily attributed to the lightweight nature of polymersomes rather than the asymmetrical structure of gold nanoparticles?

We appreciate the reviewer's input regarding the potential impact of the nanomotor weight on their speed. However, we maintain the perspective that the weight of nanomotors does not significantly influence their terminal speed. Instead, the weight is likely to affect only the rate at which nanomotors can reach their terminal/equilibrium speed based on Newton's second law. According to Stokes' law, the terminal velocity of a nanomotor is predominantly determined by the fluid viscosity, the size of the nanomotor and, more importantly, the propulsion force. As for the size of the nanomotor, a key advantage of our Au-stomatocyte system is its monodispersity, as cryo-TEM results show no stomatocyte aggregation and large Au agglomerates after Au coating can be detected. Regarding the propulsion force, as detailed in the response to Comment 2, it only requires ~ 0.3 pN for our nanomotors to achieve a speed of $125 \mu\text{m s}^{-1}$. Therefore, the asymmetrically distributed gold nanoparticles along the axial direction of the stomatocytes can induce the well-defined temperature gradient, which is capable of generating a sub-pN driven force to propel the motion of nanomotors with an ultrafast speed.

We also noted that the bottle-like nanomotors mentioned in the comment are bubble-driven nanomotors. It is crucial to emphasize that light-driven self-thermophoretic nanomotors generally exhibit greater velocities than bubble-driven nanomotors, probably due to the fundamental difference in their propulsion mechanisms. To quantitatively understand the significant variation in motion speed resulting from different propulsion forces, we compared the velocity of our Au-stomatocytes (~ 6000 Au nanoparticles, $\sim 10^{-14}$ g Au in total) in this study with that of previously reported bubble-propelled stomatocytes with manganese dioxide (MnO_2) encapsulated in their nanocavities ($\sim 10^{-14}$ g MnO_2 , *Nano Lett.* **2020**, *20*, 4472-4480). In both studies, the stomatocytes consist of the same material (PEG-PDLLA), have a similar size and the same order of magnitude of total weight but employ different propulsion mechanism. These bubble-propelled nanomotors show a slower motion performance.

Consequently, we believe that the asymmetric distribution of the gold nanoparticles is the main reason for the record-breaking speed, instead of the lightweight nature of polymersomes.

4. Similarly, the authors have not provided an explanation for the causes of this uneven distribution of gold particles.

We acknowledge the reviewer's request to elucidate the underlying causes of the uneven distribution of gold nanoparticles, as revealed by our cryo-ET analysis. The spatial distribution of Au nanoparticles is primarily governed by the morphological characteristics of the nanomotor. This template based strategy has been widely used to construct materials with precisely controlled hierarchical structures but has not been much explored for constructing asymmetry in the design of nanomotors (*Adv. Intell. Syst.* **2023**, *5*, 2200429.). To prove the template-dependent distribution of gold nanoparticles, we have also decorated PEG-PDLLA nanotubes with Au nanoparticles (see below cryo-TEM images).

Cryo-TEM images of PEG-PDLLA polymeric nanotubes and Au-tubes

With the gold nanoparticles-coated PEG-PDLLA spherical polymersomes and stomatocytes, we demonstrated that Au nanoparticles can be selectively attached to their outer surface, and their distribution relies on the morphology of the polymersomes. The stomatocyte's structure inherently possesses morphological asymmetry along its Z axis due to the presence of a unilateral cavity opening. As a result, the distribution of Au nanoparticles in this area at the same Z height will be different than that of the stomatocyte's bottom, leading to the asymmetric distribution of Au nanoparticles along the Z direction (**Fig. 5** and **Fig. S13** (also see in the attached figure)). In contrast, for the isotropic spherical polymersomes, without the presence of the cavity opening, the distribution of Au nanoparticles on the surface is isotropic across all symmetries, leading to the absence of directional motion behaviour as shown in Fig. 4 in our manuscript.

Fig. S13. Distribution of Au nanoparticles around the neck and bottom of the stomatocyte. **a)** Central cross section of an Au stomatocyte from cryo-ET, where no Au nanoparticles are present in the narrowest part of the neck. **b)** 3D position and volume map of Au nanoparticles in the neighborhood of the opening area. **c)** Volume rendering showing the cavity opening. **d-e)** Cross section showing the Au distribution in the area of the cavity opening and the stomatocyte bottom. Scale bars: **a)** 200 nm. **c-e)** 100 nm.

To further investigate the uneven distribution of gold nanoparticles on the stomatocytes, we further analysed the Au distribution specifically near and inside the stomatocyte opening (neck), based on our cryo-ET data. While there is a slight accumulation of Au nanoparticles around the opening edge, the average Au density in the entire cavity opening area ($\sim 8.5 \times 10^3 \text{ N } \mu\text{m}^{-2}$) is still lower than that of the stomatocyte's bottom ($\sim 1.9 \times 10^4 \text{ N } \mu\text{m}^{-2}$) (**Fig. 5c-h**). We speculate that during Au

deposition, the membrane in the neck area has limited access to the Au precursor due to the confined space inside the cavity compared to the membrane outside the stomatocyte, resulting in a lower formation of Au nanoparticles in the neck area. This hypothesis is supported by the fact that only 20 Au nanoparticles are present in the cavity and < 30 Au nanoparticles in the narrowest part of the neck. It is important to acknowledge that a further analysis of the results presented in the previous version of the manuscript has led to the conclusion that the highest density of Au nanoparticles is at the stomatocyte's bottom rather than in the neck region. Specifically, we realized that the boundary conditions had not been taken into account when calculating the Au density near the cavity opening of the stomatocyte. After discussion with the colleague who carried out the FEM simulation, we realized that the importance of considering the boundary conditions when estimating the surface area of each segment of the stomatocyte. In particular, when calculating the Au density for segment #40, which includes the cavity opening, it is essential to account not only for the lateral surface area but also for the contribution of the top surface area around the cavity opening. Neglecting this contribution would lead to an overestimation of the Au density around the cavity opening area. Therefore, we recalculated the lateral surface area of all 40 segments, each with a reduced segment thickness of ~ 12 nm (Fig. S14c, also see in the attached figure), and separately estimated the top surface area as well as the Au density around the stomatocyte cavity opening. The original data are plotted in Fig. 5m and the averaged Au density distribution from five data points is shown in Fig. S14d. Our revised results consistently demonstrate that the Au density at the bottom of the stomatocyte is higher than that at the opening side.

Fig. S14. Quantification of lateral surface area and Au density changes along the z-axis. **a-b)** 3D Au distribution on the surface of a single Au-stomatocyte viewed from two different directions: top view (a) and side view (b). Note that all the Au nanoparticles inside the cavity and on the top

surface around the cavity opening are removed in (a) and (b). **c)** Lateral surface area changes as a function of time. **d)** Au density changes as a function of time.

Reviewer #3

Wang et al. report on the synthesis and dynamics of light activated nanomotors in their manuscript entitled “Ultrafast Light-activated Polymeric Nanomotors”. The authors thereby target the development of a novel active particle that is made from biodegradable polymers showing a bowl shape reminiscent of stomatocytes. The polymersomes are decorated with gold nanoparticles of about 5 nm diameter, which makes them subject to photothermal heating. This optical induced heat generation is suggested to yield a very fast motion of these particles, which is determined to be around 120 $\mu\text{m/s}$. The study therefore belongs to the highly interesting field of active particles, which is currently intensely explored also with the aim of finding new smaller motors that can provide new functions.

We thank the reviewer for the clear summary of our work.

The paper is well written explaining the main results on the dynamics of these particles and also extensions which include the invasion of these particles into cells. The synthesis and characterization by cryo-electron microscopy is impressive.

We appreciate the reviewer for the positive assessment of our work.

1. The authors present data on the size of the particles, which is around 400-600 nm depending on if they are decorated with AuNP or not. It might be academic but I would count that as nanomotors. For me this is a particle with dimensions of less than 100 nm. Also, why is the size of the particles larger when decorated?

In literature, nanomotors are usually defined as small devices or particles with autonomous motion ranging from a few nanometers to several hundred nanometers. (*ACS Nano* **2019**, *13*, 11996-12005; *Nat. Commun.* **2019**, *10*, 966; *Sci. Adv.* **2020**, *6*, eabc3726.). Our Au-stomatocytes (decorating with Au NPs) with a particle size of 400-600 nm and a neck size of ~ 100 nm fall within the nanomotor catalogue and could therefore be considered as nanomotors.

Regarding nanomotor size changes, our dynamic light scattering (DLS) results indeed indicate a 20% increase in hydrodynamic size after decorating the stomatocyte with Au NPs (**Fig. 2c**). However, this doesn't necessarily imply a significant increase in the actual particle size of the stomatocyte, as the cryo-TEM image didn't show significant increase in particle size. This is due to the fact that DLS also takes into account the thickness of the hydrophilic ligands (PAA in our case) on the Au NP surface and the diffusion layer, whereas cryo-TEM does not. As a result, it is common for the hydrodynamic diameter to be larger than the size measured by TEM. To avoid confusion, we have added additional text to the main text on **page 6-7**:

...the hydrodynamic size and PDI of the Au-stomatocytes were larger than the stomatocytes (**Fig. 2c**).

2. Particles of 500 nm diameter possess a rotational diffusion time of about 100 ms in water. This rotational diffusion time separates the short time limit of the mean squared displacement model used from the enhanced diffusive limit. In other words, it separates the parabolic part of the MSD from the linear part. This is because the propulsion direction is typically connected to the geometry of the particle and rotates with the particle. Yet, there is no diffusive part visible from all the plots. Why is that? The absence of rotational diffusion points at additional forces that are relevant for the motion and raise doubts about the suggested propulsion mechanism. Also, why are the particles traveling under a specific angle to the light source?

Thank you for the reviewer's constructive comments. Rotational diffusion is a concept that describes how nanoparticles undergo random rotations due to the thermal energy of the surrounding fluid. According to the Stokes-Einstein equation, rotational diffusion is related to the size and shape of the particles, as well as the properties of the surrounding fluid. In the case of mean squared displacement (MSD), both translational diffusion and rotational diffusion play a role, specifically as $MSD(t) = MSD_{trans}(t) + MSD_{rot}(t)$. While Au-stomatocytes are expected to exhibit rapid self-rotation (approximately 0.05 s), with rotational diffusion calculated by $D_r = \tau_r^{-1} = \frac{TK_B}{8\pi\eta R^3}$, where τ_r is the reorientation time, the MSD curves display a parabolic fit at high laser power (and $\Delta t > \tau_r$). This behavior has also been observed previously (*Nano Lett.* **2020**, *20*, 4472-4480.). In addition, when calculating rotational diffusion, particle shape is often assumed to be spherical. However, the actual morphology of Au-stomatocytes is not spherical. Therefore, we hypothesize that this phenomenon is also attributed to the unique morphology of the stomatocyte and the light-propelled mechanism that counteracts the effects of rapid τ_r . This is the reason why no diffusive part is visible in all the MSD plots in this work.

Phototaxis is a common phenomenon among light-propelled micro/nanomotors, as described in various references. (*Nat. Nanotech.* **2016**, *11*, 1087–1092; *Nat. Commun.* **2021**, *12*, 2077; *J. Am. Chem. Soc.* **2022**, *144*, 3892–3901; *Adv. Healthcare Mater.* **2023**, *12*, 2301645; *J. Am. Chem. Soc.* **2016**, *138*, 6492–6497; *Nat. Commun.* **2016**, *7*, 12828.) Most of their driving forces originate from the photothermal heating-induced temperature gradient, allowing them to move from the cold side to the warm side. As a result, most light-propelled micro/nanomotors exhibit negative phototaxis, and their motion direction can be easily manipulated in response to the angle of the incident laser beam. In this work, Au-stomatocyte nanomotors also exhibit negative phototaxis because all of them are moving away from the laser source, as confirmed by tracking their motion trajectories via the Single Particle Tracking System / NanoSight. Leveraging this property, we could manipulate the motion direction of Au-stomatocyte nanomotors by adjusting the incident pathway of laser irradiation (**Video S4-5**).

3. Even though the authors report that the measurements of the particle dynamics were done with a NanoSight tracker, the overall geometry and observation is completely unclear. None would be able to reproduce the experiments except having a NanoSight tracker with exactly the same configuration then.

Details regarding the motion characterization via NanoSight can be found in the Methods section in Supplementary Information, which includes the experimental setup and data analysis. To enhance clarity, a schematic illustration detailing the experimental set-up has been included in

the Supplementary Information as **Scheme S3**. Importantly, the motion of Au-stomatocyte nanomotors can also be monitored by using a second method, namely two-photon confocal laser scanning microscopy (TP-CLSM). The corresponding videos are listed as **Video S7-S8** in the Supplementary Information as well.

4. Why did the authors choose 660 nm for photothermal heating? 5 nm AuNP have commonly their plasmonic absorption maximum at about 520 nm.

As the reviewer pointed out, the maximum absorption peak of 5 nm AuNP is around 520 nm. Therefore, selecting a 520 nm laser for photothermal performance and motion tests would yield the highest photothermal conversion efficiency and thus the best movement behaviors. However, our goal is to use nanomotors as cargo carriers for biomedical applications, such as drug delivery. In addition to considering photothermal conversion performance and movement speed, we also need to choose a suitable external laser source for subsequent biomedical applications. The wavelength of 660 nm falls within the NIR-I window (650 nm-900 nm), allowing for deeper tissue penetration and fewer side effects compared to shorter-wavelength light. Moreover, 660 nm laser is one of the most mature and widely used external light sources in clinical diagnosis and treatment (*Nat. Commun.* **2019**, *10*, 2412; *Cancer Nanotechnol.* **2023**, *14*, 67; *Chem. Soc. Rev.* **2019**, *48*, 2053-2108.). Therefore, considering these factors, we selected a 660 nm external laser as laser source for this work.

5. The authors mention a power of 1.5 W that is used for the heating and propulsion experiments. Power is only useful when providing the illuminated area as well.

We acknowledge the reviewer's suggestion and have calculated the power intensity. We would like to point out that it is also common in literature to use the displayed power value of the laser device (e.g. *Nat. Nanotech.* **2018**, *13*, 304-308; *Adv. Healthcare Mater.* **2023**, *12*, 2203018; *Mater. Today Chem.* **2023**, *30*, 101533.). However, we agree with the reviewer that it's important to show the power density as well. Consequently, we have added a supplementary **Table S4** in the **Supplementary Information** to list the output power values and their corresponding power densities in this work (**Page S9**). The power value of 1.5 W corresponds to a power density of ~ 3000 mW cm⁻².

6. The authors report the absorption spectrum in Fig. 2d of the main text. Given the shape, I would rather think that the authors measured extinction, e.g., the removal of light from the incident direction, which also contains scattering.

The UV spectrometer specifically records the excitation spectrum rather than the absorption spectrum. However, in dilute solutions, we can ignore the influence of scattering, including Rayleigh Scattering, Mie Scattering, Tyndall Scattering, and others. In this work, we diluted the samples before UV spectrometer measurements, ensuring that the recorded excitation spectrum was approximately equal to the absorption spectrum. In many related studies, the extinction spectrum is often simplified to the absorption spectrum (*J. Am. Chem. Soc.* **2023**, *145*, 288-299; *Adv. Mater.* **2017**, *29*, 1604894; *Nat. Commun.* **2020**, *11*, 1724.).

7. Figure 3 seems to report the local average temperature of the sample as the IR images. Is the temperature observed there really corresponding to the temperature at the stomatocyte surface? The IR images only show the solution temperature, not the stomatocyte surface temperature. Our observation of a local average temperature is due to the fact that the area illuminated by the laser has the highest temperature, from which the heat generated subsequently diffuses throughout the solution, creating a temperature gradient. This is a common phenomenon that has already been observed before (*ACS Nano* **2023**, *17*, 16089-16106.).

Minor stuff:

1. The authors mention that the particles have a „shape of the stomatocytes leads to a lower drag coefficient that could be beneficial for their movement“. In fact active particles do not propel against friction and the shape is of other importance than for particles, which are dragged through the liquid.

We thank the reviewer for the comment and agree that the statement in question may lead to a potential misunderstanding. It's important to clarify that the shape of the stomatocyte itself does not inherently result in a lower drag coefficient. In fact, the actual drag coefficient depends on the specific orientation of the active Au-coated stomatocyte as it moves under laser irradiation (*Aerosol Sci. Technol.* **1987**, *6*, 153-161). Through drag force calculations, we found that to overcome drag forces at a speed of $125 \mu\text{m s}^{-1}$, it requires a propulsion force of ~ 0.3 pN for our nanomotors, considering the solution of the full Navier-Stokes equations and taking into account the shape factor of the stomatocyte (**Fig. S16**). In contrast, spherical polymersomes, according to Stokes' law, would require ~ 0.5 pN. This observation indicates that the geometric shape of our nanomotor provides an advantage over a completely spherical shape during movement. However, we believe that the shape-dependent drag coefficient is not the primary determinant of the motility of these nanomotors. Instead, the well-defined temperature gradient induced by the uneven spatial distribution of Au nanoparticles on the surface of the stomatocyte plays a more important role in governing the motion behavior of these nanomotors, as we confirmed through the FEM simulations (**Fig. S15**).

2. The authors also mention that the particles cross the cell membrane due to their kinetic energy. Also this is not the correct physical picture.

We agree with this reviewer that the statement may be inaccurate and could lead to potential misunderstanding. Therefore, we have replaced the “kinetic energy” with “mechanical disruption”.

3. Temperatures are commonly given in Kelvin or degree centigrade. Temperature difference are given in Kelvin.

We appreciate the reviewer's careful attention to the manuscript, and we agree with the suggestion. We have corrected the temperature difference unit to Kelvin.

4. The authors refer to Golestanian self-diffusiophoretic model. Actually this model is a general

description of the mean squared displacement of a self-propelled Brownian particle and not bound to diffusiophoresis.

The Golestanian self-diffusiophoretic model has been widely used in the study of self-propelled Brownian particles and diffusiophoretic particles, as evidenced by previous publications (*Sci. Adv.* **2023**, *9*, eabg3015; *J. Am. Chem. Soc.* **2022**, *144*, 1634-1646; *Phys. Rev. Lett.* **2010**, *105*, 268302; *Phys. Fluids* **2016**, *28*, 053107.). However, we acknowledge the reviewer's suggestion that this model still requires improvement for characterizing diffusiophoresis-based micro/nanomotors.

5. The authors report the surface charge of the stomatocytes before and after the AuNP deposition, but without giving a unit.

We appreciate the careful attention to our manuscript. We have added the unit for surface charge (mV) to **Table S3** in the Supplementary Information.

6. The axis labels in Figure 4c is unreadable and Figure 4 is in general of low quality with different font sizes everywhere. Also giving just X-position and Y-position as axis labels in Fig. 4G is not sufficient for a publication submitted to Nature Communications.

We have standardized the font size of the figures and enhanced the quality of all the figures accordingly.

REVIEWER COMMENTS

Reviewer #2 (Remarks to the Author):

The author has responded to all the raised questions, we believe that now the work is ok for publication.

[Note from the Editor: Reviewer #2 was asked to look also over the response given to Reviewer #1]

Reviewer #3 (Remarks to the Author):

I would like to thank the authors for the detailed response. I appreciate the effort to clarify the my questions and comments to the manuscript. As mentioned before, I appreciate the results of the preparation and cryo-TEM. Yet, a number of issues still remain after the detailed response.

Concerning my question #2:

Self-propelled microscopic objects can only move due to some shape asymmetry, like Janus particles due to the time symmetry of hydrodynamic equations at small Reynolds numbers. Therefore, the propulsion direction is bound to the geometry and the propulsion direction randomises due to the rotational diffusion. This is why rotation enters the mean squared displacement, yet not in the given additive way that the authors describe in their response. (The described additivity of the translational and rotational MSD is incorrect.) This is the reason, why beyond the rotational diffusion time, the motion of the objects has to be enhanced diffusive with a linear MSD. Only at shorter times than the rotational diffusion times the MSD can be advective. If this is not the case, then either the objects are heated asymmetrically or pushed by radiation pressure, where in the latter case this is just an external force and not self-propulsion. Thus, the source of the observed dynamics must be clarified before publication.

Concerning my question #6:

According to my knowledge, an UV spectrometer is measuring absorption or extinction and not an excitation spectrum, which would require fluorescence detection. I was wondering, why the corresponding spectra displayed show a dependence of the "absorbance" on the wavelength, which is reminiscent of the Rayleigh spectrum of small particles and proportional to $1/\lambda^4$. This seems to be also prominent for the Stomatocytes.

Concerning my question #7:

If the solution temperature is up to 50 degree celsius, then this would mean based on heat conduction that the gold particles should be substantially hotter. Can the authors comment on that

and even discuss the temperature quantitatively based on the excellent TEM images giving the density of Au nanoparticles on the surface. This should be possible.

Minor stuff question #1:

My question was targeting the propulsion mechanism and its relation to the drag coefficient. In fact self propelled particles do not propel against the drag of the liquid. They propel because of the drag and exhibit a force balance between the viscous friction and the stresses self-generated in the liquid. In case the objects are not self-propelled but externally propelled by some force like radiation pressure, the drag force would be of relevance. I think this needs a clarification of the propulsion mechanism.

So overall my main concerns regarding the discussed dynamics persists and I cannot recommend the paper for publication in Nature Communications.

Please find below a point-by-point response to the referee's comments.

Reviewer #3

I would like to thank the authors for the detailed response. I appreciate the effort to clarify the my questions and comments to the manuscript. As mentioned before, I appreciate the results of the preparation and cryo-TEM. Yet, a number of issues still remain after the detailed response.

We would like to thank the reviewer for their time and questions.

Concerning my question #2:

Self-propelled microscopic objects can only move due to some shape asymmetry, like Janus particles due to the time symmetry of hydrodynamic equations at small Reynolds numbers. Therefore, the propulsion direction is bound to the geometry and the propulsion direction randomises due to the rotational diffusion. This is why rotation enters the mean squared displacement, yet not in the given additive way that the authors describe in their response. (The described additivity of the translational and rotational MSD is incorrect.) This is the reason, why beyond the rotational diffusion time, the motion of the objects has to be enhanced diffusive with a linear MSD. Only at shorter times than the rotational diffusion times the MSD can be advective. If this is not the case, then either the objects are heated asymmetrically or pushed by radiation pressure, where in the latter case this is just an external force and not self-propulsion. Thus, the source of the observed dynamics must be clarified before publication.

Response:

We would like to thank the reviewer for their interesting comment. First of all, our particles also have a clear shape asymmetry and a heterogeneous distribution of the gold nanoparticles, with a higher density at the bottom of the stomatocyte structure (see cryo-EM and quantitative analysis results (**Fig.5** and **Fig.S11**)). Although this is not a Janus type particle, it has similar characteristics to allow for motility to be observed.

We agree with the reviewer that our nanomotors exhibit a parabolic MSD fit at $\Delta t > \tau_r$ ($\tau_r = 0.05$ s according to $D_r = \tau_r^{-1} = \frac{TK_B}{8\pi\eta R^3}$).

Generally, the nanoscale motor's rotational diffusion is typically ignored in the calculation of mean square displacement as the rotational diffusion time is very small - this has been previously highlighted in the literature (See e.g.: *Angew. Chem. Int. Ed.* **2018**, 57, 6838–6842; *ACS Nano* **2021**, 15, 14218–14228; *Adv. Funct. Mater.* **2024**, 2314568; *Small* **2020**, 16, 2003834; *Nat Commun.* **2021**, 12, 2077). Translational diffusion occurs over larger length scales (compared to rotational diffusion), leading to particle motion across longer distances. In our case, the nanomotor motion is induced by thermal gradients within the solution, spanning micron-scale distances between regions of varying temperatures (higher and lower temperatures). As we observe motion of our particles at the microscopic level, the predominant influence we detect is therefore translational diffusion. On the other hand, rotational diffusion operates at the nanoscale and its effects are not directly observable under our measurement conditions. Therefore, in our opinion, it is indeed safe to disregard the effects of rotational diffusion in our MSD calculations.

Moreover, in line with the reviewer's statement on particle asymmetry, indeed our particles are heated asymmetrically owing to the heterogeneous distribution of the Au NPs. This asymmetry results in the formation of a temperature gradient surrounding the particle, originating from both particle heating and subsequent dissipation of heat – this is supported by theoretical simulations (See: **Fig. S15**). Therefore, the temperature distribution in the solution is heterogeneous, especially surrounding the particle. Such heterogeneity introduces a temperature gradient. Logically, regions with higher temperature result in more thermal motion of the stomatocytes, which, in turn, impact both translational (D_t) and rotational (D_r) diffusion. The values for the diffusion coefficients can be calculated. By using the Stokes-Einstein-Debye equation $D_r = k_B T / 8\pi\eta r^3$, $D_r = 1.65 \times 10^{-11} \text{ rad}^2 \text{ s}^{-1}$ ($r = 235 \text{ nm}$ at 50°C , the average solution temperature due to laser irradiation). By using the Stokes-Einstein equation $D_t = k_B T / 6\pi\eta r$, $D_t = 1.31 \times 10^{-15} \text{ m}^2 \text{ s}^{-1}$ ($r = 235$ at 50°C , the average solution temperature due to laser irradiation). In a heterogeneous environment, using Arrhenius equations for translational and rotational diffusion we can calculate the translational and rotational activation energy, respectively

$$\text{Translation diffusion: } D_t = D_{t0} \exp(-E_{ta}/K_B T)$$

$$\text{Rotational diffusion: } D_r = D_{r0} \exp(-E_{ra}/K_B T)$$

Whereas D_0 is the pre-exponential factor, E_a is the activation energy. Typically, D_{t0} and D_{r0} in liquids fall in the range of 10^{-9} to $10^{-11} \text{ m}^2 \text{ s}^{-1}$ and 10^8 to $10^9 \text{ rad}^2 \text{ s}^{-1}$.

Therefore, E_{ta} and E_{ra} are approximately $4.73 \times 10^{-20} \text{ (J/mol)}$ and $1.92 \times 10^{-19} \text{ (J/mol)}$

Consequently, E_{ta} for translational diffusion is significantly smaller than E_{ra} . This difference is due to the consistent heterogeneous interactions between particles and their surroundings. Lower E_a for translational diffusion implies that our nanomotor can more easily undergo translational motion upon irradiation, compared to rotational diffusion.

Extra control experiments with non-Au coated particles demonstrated that such particles do not move, regardless of the laser power (See: Fig. 4b) as they do not heat up and do not create the heterogeneous environment necessary for thermophoretic motion. We therefore think that we can state that our particles exhibit self-propulsion, propelled by the mechanism of thermophoresis.

Concerning my question #6:

According to my knowledge, an UV spectrometer is measuring absorption or extinction and not an excitation spectrum, which would require fluorescence detection. I was wondering, why the corresponding spectra displayed show a dependence of the "absorbance" on the wavelength, which is reminiscent of the Rayleigh spectrum of small particles and proportional to $1/\lambda^4$. This seems to be also prominent for the Stomatocytes.

Response:

We agree with this reviewer that the UV-vis spectrometer is an instrument to measure absorption or extinction. Nevertheless, in the context of solutions containing metallic nanoparticles, UV-vis spectroscopy serves as a standard characterization technique (See *Anal. Chem.* **2019**, 91, 14639-14648; *Proc. Natl. Acad. Sci. USA*, **2017**, 114, E3110-E3118, and *Nat. Commun.* **2018**, 9, 1074). In our study, the primary aim of employing UV-vis spectra was to verify the presence of gold nanoparticles on the surface of stomatocytes. Furthermore, to ensure clarity and align with previous publications, we have revised the terminology to

refer to extinction spectra, consistent with the terminology used in *Adv. Mater.* **2021**, 33, 2008540, and *Nat. Commun.* **2022**, 13, 3064.

Concerning my question #7:

If the solution temperature is up to 50 degree Celsius, then this would mean based on heat conduction that the gold particles should be substantially hotter. Can the authors comment on that and even discuss the temperature quantitatively based on the excellent TEM images giving the density of Au nanoparticles on the surface. This should be possible.

Response:

In response to the inquiry posed by the reviewer, we conducted several calculations to approximate the thermal field within the aqueous solution. Our initial task was to quantitatively replicate the heating curve and the thermal distribution inside the aqueous tube, which was subjected to constant laser radiation heating at 1.5 W with a wavelength of 660 nm. Throughout the heating process, the aqueous solution was warmed by laser radiation, while concurrently being cooled by the surrounding air through natural convection. Hence, to accurately model the heating curve, it was imperative to ascertain both the energy absorbed by the Au nanoparticles under laser radiation and the rate of heat release from the aqueous tube. Unfortunately, comprehensive data for these parameters were not readily available, necessitating several preliminary estimations. For instance, we estimated that 1 mg of the aqueous solution contained approximately 1×10^{11} nanomotors, and according to transmission electron microscopy (TEM) analysis, each nanomotor comprised about 5×10^3 gold nanoparticles. Based on Mie theory, a 5 nm spherical nanoparticle is capable of absorbing 4.1×10^{-15} W of energy under our experimental conditions. Armed with this information, we calculated that the heat generation within a 1 mg of the aqueous solution would be approximately 2.05 W.

Subsequently, we estimated the rate of heat transfer between the aqueous tube and ambient air. This heat flux was evaluated using Newton's law of cooling, expressed as $q_c = \alpha(T_{\text{aqueous}} - T_{\text{air}})$, where α denotes the heat transfer coefficient. Precisely estimating α proved challenging due to its dependency on various environmental factors such as humidity and wind speed in the laboratory. Nevertheless, based on our experience, we anticipated that α would be in the order of $o(10)$ for natural convection in a typical indoor environment.

Based on these estimations, we developed a numerical model to simulate heat transfer within the aqueous tube, meticulously aligning the tube's geometry and the aqueous loading volume with the experimental setup. The heat absorption by the Au particles was modeled as a volumetric heat source, initially set to 2.05 W/mg and later adjusted to 0.5 W/mg. To align with the observed experimental heating curve, we fine-tuned the heat transfer coefficient, ultimately fixing it at $\alpha = 35 \text{ W/m}^2\cdot\text{K}$. After numerous iterations, we successfully aligned the modeled thermal field with the experimental measurements, as depicted in below image (Fig Rev 1).

Fig Rev 1. a) Scheme of the simulation process, b) Temperature effects caused by laser irradiation (660 nm, 1.5 W) of Au-stomatocytes, experimental results and simulation

Gold nanoparticles do not need to sustain an elevated temperature to effectively transfer heat to the surrounding fluid. This assertion can be straightforwardly substantiated by applying Fourier's Law of Heat Conduction. First, we calculated the Biot number (Bi) for a gold nanoparticle, finding $Bi = \alpha \cdot r / \lambda$ to be approximately 1×10^{-11} , significantly less than one. This small Biot number allows us to infer that the temperature distribution within a gold nanoparticle is uniform.

The heat absorption by a gold nanoparticle is quantified as $Q = 4.1 \times 10^{-15}$ W. Given a spherical Au particle, its surface area A is calculated as $4\pi r^2$, which equals 78.5 nm^2 . The thermal conductivity of the surrounding liquid is assumed to be about 0.6 W/mK . Additionally, we consider a heat conduction length scale, δ , of 100 nm . By applying Fourier's Law, we ascertain that the temperature differential between the Au nanoparticle and its surrounding liquid, ΔT , equates to $Q / (A \cdot \lambda) \cdot \delta$, resulting in a negligible value of $5.2 \times 10^{-6} \text{ }^\circ\text{C}$.

We also conducted a Finite Element Method (FEM) analysis to visualize the thermal field around a single nanoparticle (Fig Rev 2). Under the specified heating conditions, the temperature disparity between the nanoparticle and the surrounding fluid is minimal. Consequently, it can be deduced that the temperature within the aqueous solution remains homogeneous throughout the heating process. Despite acting as a heat source, the nanoparticles maintain a temperature nearly identical to that of the surrounding liquid, underscoring the efficiency of heat transfer without the need for a significant temperature gradient.

Fig Rev 2. FEM analysis of equilibrium temperature distribution surrounding a single nanoparticle immersed in a liquid.

Minor stuff question #1:

My question was targeting the propulsion mechanism and its relation to the drag coefficient. In fact self-propelled particles do not propel against the drag of the liquid. They propel because of the drag and exhibit a force balance between the viscous friction and the stresses self-generated in the liquid. In case the objects are not self-propelled but externally propelled by some force like radiation pressure, the drag force would be of relevance. I think this needs a clarification of the propulsion mechanism.

Response:

In line with our answer to the reviewer's first question and cryo-EM results, the gold nanoparticles are unevenly distributed on the surface of stomatocytes, resulting in a temperature gradient upon laser irradiation – this is evident by theoretical simulations (see: **Fig. 5**, and **Fig. S11-Fig. S15**). The motility of our particles is therefore driven by thermophoresis, as evident by control experiments where Au uncoated particles underwent typical Brownian motion. Therefore, it is safe to disregard that motion is a result of radiation pressure or drag. Thermophoresis-driven propulsion of Au coated particles has been demonstrated before in the literature (See: *Angew. Chem. Int. Ed.* **2020**, 59, 14368–14372, *J. Am. Chem. Soc.* **2016**, 138, 20, 6492–6497, *J. Am. Chem. Soc.* **2022**, 144, 25, 11246–11252, *Angew. Chem. Int. Ed.* **2020**, 59, 16918–16925, and *J. Am. Chem. Soc.* **2022**, 144, 9, 3892–3901). In the main text, we have confirmed that the propulsion mechanism of our nanomotors is thermophoresis based on the simulated temperature distribution results (**Fig. 5o**), see the discussion on **Page 13-14**. We also considered the effect of radiation pressure; therefore we performed a series of controls, in which we irradiated pure stomatocytes, Au-polymersomes, and 300 nm spherical Au NPs. Based on the lack of either a motile agent or a clear asymmetric structure, we would expect these particles not to show motile behavior, which was indeed observed (Fig Rev 3).

Fig Rev 3 MSD curves (top left, **Fig. 4b**) and trajectories (**Fig. 4c**) of pure stomatocytes (top right), Au-polymersomes (bottom left), and 300 nm Au NPs (bottom right) upon laser irradiation (660 nm, 1W).

With best regards

Jan van Hest

REVIEWERS' COMMENTS

Reviewer #2 (Remarks to the Author):

The authors have responded to all the questions, the manuscript is now ready for publication.

Reviewer #4 (Remarks to the Author):

This is a very interesting manuscript since the fabrication materials of nanomotors and fuel-free propulsion are biodegradable or biocompatible. The synthesis and characterization are done well, and also the motion analysis is performed well. I have similar questions with other referees, and now authors have responded them very well. In view of this, I am pleased to recommend it to be accepted for publication as it is.